# Spermidine and 1,3-Diaminopropane Have Opposite Effects on the Final Stage of Cephalosporin C Biosynthesis in High-Yielding *Acremonium chrysogenum* Strain

**DOI:** 10.3390/ijms232314625

**Published:** 2022-11-23

**Authors:** Alexander A. Zhgun, Mikhail A. Eldarov

**Affiliations:** Group of Fungal Genetic Engineering, Federal Research Center “Fundamentals of Biotechnology” of the Russian Academy of Sciences, Leninsky Prosp. 33-2, 119071 Moscow, Russia

**Keywords:** filamentous fungi, *Acremonium chrysogenum*, biosynthesis of secondary metabolites, cephalosporin C, polyamines, spermidine, acetyl coenzyme A

## Abstract

The addition of exogenous polyamines increases the production of antibiotic cephalosporin C (CPC) in *Acremonium chrysogenum* high-yielding (HY) strain during fermentation on a complex medium. However, the molecular basis of this phenomenon is still unknown. In the current study, we developed a special synthetic medium on which we revealed the opposite effect of polyamines. The addition of 1,3-diaminopropane resulted in an increase in the yield of CPC by 12–15%. However, the addition of spermidine resulted in a decrease in the yield of CPC by 14–15% and accumulation of its metabolic pathway precursor, deacetylcephalosporin C (DAC); the total amount of cephems (DAC and CPC) was the same as after the addition of DAP. This indicates that spermidine, but not 1,3-diaminopropane, affects the final stage of CPC biosynthesis, associated with the acetylation of its precursor. In both cases, upregulation of biosynthetic genes from beta-lactam BGCs occurred at the same level as compared to the control; expression of transport genes was at the control level. The opposite effect may be due to the fact that N^1^-acetylation is much more efficient during spermidine catabolism than for 1,3-diaminopropane. The addition of spermidine, but not 1,3-diaminopropane, depleted the pool of acetyl coenzyme A by more than two-fold compared to control, which could lead to the accumulation of DAC.

## 1. Introduction

Filamentous fungi are one of the most important sources for obtaining pharmaceutically significant secondary metabolites (SMs), such as antibiotics, statins, and immunosuppressants [1,2]. Currently, the main method for obtaining high-yielding (HY) producers of SMs in fungi is the so-called classical strain improvement (CSI), associated with screening after multi-round random mutagenesis by X-ray, ultraviolet radiation, and chemicals such as chlormethine, nitrogen mustard, and nitrosoguanidine [3,4]. The application of CSI technology to various strains of filamentous fungi since the 1950s has increased their production of target SM by 100–1000 or more times compared to the original wild-type (WT) strains [3,5,6,7,8]. It turns out that CSI has its technological limit; in the process of multi-round mutagenesis, there comes a stage when the next mutagenic action no longer leads to an increase in the yield of the target SMs [4]. However, it was found recently that the addition of aliphatic polyamines (PAs) during the fermentation of HY fungal strains (which reached their technological limit of improvement by classical methods) can additionally increase the yield of target SMs by 15–45% [9,10,11].

Low-molecular-weight aliphatic PAs such as 1,3-diaminopropane (1,3-DAP), putrescine, spermidine (SPD), and spermine are widely distributed in nature [12]. They are found in most cells of eukaryotes, bacteria, and archaea, as well as in viral particles [13]. PAs perform various functions in various organisms, most of which are associated with cell growth, proliferation, protection of the cell from stress, regulation of ion channels, and regulation at the level of transcription and translation [12,14,15,16,17,18,19]. At the same time, the intracellular concentration of PAs itself is an important regulatory factor [20]. In this regard, there is a rigid homeostasis of the concentration of PAs in the cells of an organism [21,22]. For this, there are both enzymes of the biosynthetic pathway and enzymes of PAs catabolism. The key stages of both biosynthesis and catabolism are under strict control and regulated by a feedback mechanism based on the concentration of PAs [23,24]. For example, one of the key enzymes of PAs biosynthesis, ornithine decarboxylase (ODC, EC: 4.1.1.17), which triggers the formation of PAs from L-ornithine (removing this amino acid out of the urea cycle), is regulated at levels transcription, translation, and protein turnover [25]. Another significant PAs biosynthetic enzyme, S-adenosylmethionine decarboxylase (SAMdc, EC: 4.1.1.50), converts S-adenosyl-L-methionine (AdoMet) to decarboxylated S-adenosyl-L-methionine (dcAdoMet). SAMdc activity is also regulated by changes in the intracellular content of PAs [26]. This regulation is extremely important since AdoMet, which is used in most other cellular reactions for methylation, is excreted for the biosynthesis of PAs [27,28]. ODC activity converts L-ornithine into putrescin, which is then aminopropylated by spermidine synthase (SpdS, EC: 2.5.1.16) using dcAdoMet as a donor to form SPD. Spermidine, in turn, is aminopropylated by spermine synthase (SpmS, EC: 2.5.1.22), which also uses dcAdoMet as a donor to form spermine [29]. The key regulator of PAs content at the level of their catabolism is spermidine/spermine-N^1^-acetyltransferase (SSAT, EC: 2.3.1.57) [30]. At elevated concentrations of SPD and spermine, SSAT performs N^1^-acetylation (N^1^,N^12^-diacetylation is also possible for spermine) by adding acetyl groups from acetyl-coenzyme A (acetyl-CoA) to the aminopropyl end (s) of SPD and spermine. Acetylated derivatives, on the one hand, become available for further catabolism of PAs by enzymes such as N^1^-acetylpolyamine oxidase (APAO, EC: 1.5.3.13); on the other hand, they are readily excreted from the cell [30,31,32].

The object of our study is a high-yielding producer of the antibiotic cephalosporin C (CPC), strain *A. chrysogenum* HY (RNCM 408D), obtained as a result of the CSI program from a natural isolate, *A. chrysogenum* WT (ATCC 11550) [7]. *A. chrysogenum* is an exclusive industrial producer of CPC, which serves as a feedstock for the subsequent in vitro synthesis of cephalosporin antibiotics of the 1st–5th generations [33]. Biosynthesis of CPC in *A. chrysogenum* is carried out due to the functioning of two biosynthetic gene clusters (BGCs), into which, respectively, the so-called “early” and “late” genes for beta-lactam biosynthesis, transport, and regulation are assembled [34]. These clusters are located on different chromosomes (in wild-type strains, “early” BGC is localized on chromosome VI, and “late” BGC is localized on chromosome I) and are coordinately regulated [35]. The “early” BGC of beta-lactams contains biosynthetic genes for the “early” (*pcbAB*, *pcbC*) and “middle” (*cefD1*, *cefD2*) stages of biosynthesis, as well as genes for the transport of intermediates through cellular compartments (*cefP*, *cefM*, and *cefT*) and *cefR* for pathway-specific regulator. The “late” BGC of beta-lactams contains genes for the final stages of CPC biosynthesis (*cefEF* and *cefG*).

*A. chrysogenum* HY typically produces 9–12 g of CPC during laboratory fermentation in shake flasks, 200–300 times higher than WT strains [7]. Previous studies have shown that significant chromosomal rearrangements occurred during *A. chrysogenum* HY mutagenesis [35]. These changes, however, did not affect beta-lactam BGCs; in particular, no duplications or translocations were found [35,36]. However, there was a significant upregulation of the genes for the biosynthesis of beta-lactams by 10–350 times [36]. Along with a huge increase in the CPC production, this strain has a sharp decrease in viability, which is expressed in slow growth, a reduction in the formation of conidia [37], thinning of the cell wall [38], a decrease in plasma membrane H^+^-ATPase (PMA, EC: 7.1.2.1) activity and intracellular ATP content [39]. These data show that, as a side event of random mutagenesis, *A. chrysogenum* HY became attenuated. This HY strain also lost a characteristic yellow-cream color, which develops in the WT strain after 5–10 days, depending on the cultivation conditions on the agarized nutrient medium and due to the biosynthesis of the secondary metabolite sorbicillin [11]. In a series of developed recombinant *A. chrysogenum* HY/PMA strains with PMA activity consistently increasing to the level in *A. chrysogenum* WT strain, the ATP content and CPC yield decreased proportionally [40]. This may indicate that the decrease in PMA activity in *A. chrysogenum* HY is an adaptive change selected during the CSI program, which allowed the release of a certain amount of ATP for energy-consuming CPC biosynthesis [40].

Another phenotypic difference of the *A. chrysogenum* HY strain is associated with unexpectedly high resistance to ODC inhibitors such as α-difluoromethylornithine (DFMO, or eflornithine) and APA (1-aminooxy-3-aminopropane), which specifically kill *A. chrysogenum* WT, but not the attenuated *A. chrysogenum* HY strain [41]. It was also found that during the fermentation of *A. chrysogenum* HY, the content of PAs (SPD, spermine) was increased by 4–5 times compared to the *A. chrysogenum* WT [41]. The introduction of exogenous PAs, such as SPD or 1,3-DAP, led to an increase in the germination of colonies and morphological changes in the *A. chrysogenum* HY strain on complex agarized (CPA) medium [11]. It was also shown that the addition of 5 mM PAs during submerged fermentation on a complex (CP) medium increases CPC production in *A. chrysogenum* HY by 15–20% and upregulates the genes from beta-lactam BGCs [11]. The data obtained for *A. chrysogenum* HY: (i) increased resistance to inhibitors of PAs biosynthesis, (ii) a shift in PAs homeostasis, expressed in an increase in their cellular content, (iii) an increase in CPC production upon exogenous administration of Pas—indicate the intersection of CPC biosynthesis and PAs metabolism. However, the levels at which this crossover of metabolic pathways occurs are unknown. At the same time, understanding the molecular basis for increasing the production under the action of PAs is extremely important for the correct strategy for cultivating fungal producers since it was shown that exogenous PAs can both increase and decrease the level of target SM [42]. The available data of the PAs effect on CPC yield in *A. chrysogenum* HY were obtained on CP medium, with an indeterminate set of nutrients, where it is impossible to reveal the role of individual components and which itself contains a certain number of PAs.

In this regard, the purpose of our work was to study the effect of PAs on the production of CPC on a medium with defined and specially selected components. It is important that such a synthetic (SN) medium, on the one hand, contain a clear and limited amount of nutrients and, on the other hand, provide the necessary conditions for the growth and high production of CPC in *A. chrysogenum* HY. It turned out that 1,3-DAP and SPD act differently on such a medium; the addition of 1,3-DAP leads to an increase in CPC production, while the addition of SPD, on the contrary, reduces it. Concurrently, when SPD (but not 1,3-DAP) was added, the biosynthetic pathway precursor of CPC, deacetylcephalosporin C (DAC), was significantly accumulated. As a result, the total amount of cephems (DAC and CPC) with the addition of both PAs was very close and 12–15% higher than in the control. This indicates that on the SN medium, the addition of SPD, but not 1,3-DAP, selectively affects the last step in CPC biosynthesis, which is also known as the rate-limiting step [43]. In order to understand the molecular basis of this phenomenon, we studied the expression of genes for the biosynthesis and transport of beta-lactams after the addition of 1,3-DAP and SPD and without additives (control). However, we did not find significant differences; the addition of both PAs resulted in a similar upregulation compared to control. Importantly, there was no significant difference in the level of expression of the *cefG* gene, whose product, deacetylcephalosporin-C acetyltransferase (CefG, EC: 2.3.1.175), converts DAC to CPC. Since acetyl-CoA is also required for CefG to function, the opposite action of PAs may be due to the fact that acetyl-CoA is consumed for the catabolism of SPD and not 1,3-DAP (for which the N^1^-acetylation reaction proceeds several orders of magnitude worse). Adding SPD to a relatively nutrient-poor SN medium depletes the acetyl-CoA pool and accumulates DAC. We were able to confirm this hypothesis after measuring the content of acetyl-CoA in the samples on both media. It turned out that in the CP medium, the addition of polyamines did not change the content of cellular acetyl-CoA compared to the control, while in the SN medium, the addition of SPD, but not 1,3-DAP, led to a significant (more than two-fold) change in the content of acetyl-CoA compared to the control. Thus, the obtained data point to the crosstalk of the pathways of CPC biosynthesis and SPD catabolism at the level of the common substrate, acetyl-CoA.

In our work, we showed for the first time the opposite effect of polyamines on SMs production in improved fungal strains. This knowledge can be used both for the targeted obtaining of HY fungal producers and for changing the components of the medium in order to increase the yield of the target SM.

## 2. Results

### 2.1. Determining of Nitrogen (N) Source and Doses of PAs for Synthetic (SN) Medium

In the first stage of our work, we developed an effective SN medium for the growth of *A. chrysogenum* HY with PAs. The level of CPC production in this strain, which became attenuated after multi-round mutagenesis, is extremely sensitive to the medium composition. The CP medium developed for its fermentation contains a significant amount of nutritional components of natural origin, such as 105 g/L of corn extract, 60 g/L of corn dextrin, 20 g/L of corn starch, 20 g/L soybean oil [7]. In this regard, in the case of switching to exclusively synthetic components for the fermentation of this strain, a sharp drop in the level of CPC biosynthesis is possible, at which it will be difficult to determine the effect of PAs.

One of the most important components of the nutrient medium for filamentous fungi is the source of nitrogen (N) consumption [44,45]. Nitrogen metabolism is also critical for nitrogen-containing polyamines; there is an interplay between PAs and N [46]. It was described that PAs/N interplay connects N metabolism, carbon fixation, and secondary metabolism pathways [46,47]. In this regard, the most preferred source of N for *A. chrysogenum* HY was determined on an agarized semi-synthetic Czapek (CZA) medium, and then the most effective concentrations for introducing PAs were determined on a medium with a selected N source.

#### 2.1.1. Determining N Source for Cultivation of *A. chrysogenum* HY

It is known that various mutations, for example, in the genes of global N regulators such as *AcareA* and *AcareB*, can change the consumption of nitrogen sources and affect the biosynthesis of CPC in *A. chrysogenum* [48,49]. In particular, disruption of *AcareA* resulted in growth failure on media using NaNO_3_, uric acid, and low concentrations of NH_4_Cl, L-Gln, or urea as the sole source of nitrogen [48]; disruption of *AcareB* resulted in growth failure on media using of NH_4_Cl, L-Gln, or urea as the sole source of nitrogen [49]. Random mutagenesis in *A. chrysogenum* HY could lead to a variety of different side mutations, including disruption of the consumption of N sources.

Therefore, in order to determine possible changes in the intake of N sources, we compared the growth of this HY strain and its parental WT strain on a CZA medium, supplemented with NaNO_3_, NH_4_Cl, urea, L-Asn, or L-Gln as the sole source of N (Figure 1). The concentration range used was 1–50 mM; concentrations below 1 mM for all compounds turned out to be ineffective, and concentrations of 100 mM or more for all compounds turned out to be toxic (Appendix A). Both the type of the N source and its concentration strongly influenced colony growth and morphology (Figure 1a, Appendix A). The trend of changes for the size of colonies (in the series of 1–10–50 mM mole) was the same for both strains on media with NH_4_Cl, urea, and L-Gln (Figure 1b,c). NH_4_Cl turned out to be toxic for both strains; the size of colonies decreased in the range 1–10–50 mM. For urea, the concentration of 10 mM turned out to be the most optimal; at 50 mM, the growth of colonies decreased, probably due to the toxicity of this concentration for both strains. For L-Gln, the highest colony growth was observed at 50 mM. At the same time, in both strains, the size of colonies at 10 mM was somewhat smaller than at 1 mM. However, colonies were denser at 10 mM than at 1 mM. On agarized Czapek-Dox (CDA, with the addition of NaNO_3_) medium, *A. chrysogenum* WT had the largest colony size at 10 mM; at 50 mM the colonies were slightly smaller but denser. On CDA *A. chrysogenum* HY had very close and optimal growth in the concentration range of 10–50 mM. On agarized Czapek-L-Asn (CAA) medium, the strains showed different trends in colony size (Figure 1b,c). *A. chrysogenum* WT in CAA medium had the largest colonies at 10 mM; when the concentration was increased to 50 mM, there was a slight decrease in the size of the colonies, although the density of the colony increased. *A. chrysogenum* HY in CAA medium in the series 1–10–50 mM had a clear tendency to increase both the size and density of colonies. This difference may be due to the fact that L-Asn is better consumed by the *A. chrysogenum* HY than by the *A. chrysogenum* WT. The addition of 50 mM L-Asn resulted in the largest colonies for *A. chrysogenum* HY in this study; microscopic analysis also showed an increase in cell size (Figure 1a,c, Appendix A).

High concentrations (50 mM) of some N sources compounds were found to be toxic, which resulted in reduced growth of the wild-type strain Czapek-N medium, supplemented with NH_4_Cl, urea, or L-Asn (Figure 1b), and the high-yielding strain—on NH_4_Cl (Figure 1c). At this concentration of NH_4_Cl or urea, the production of secondary metabolite sorbicillin was also disrupted in the WT strain, which was expressed in the absence of the characteristic yellow-cream color. Both strains grew well on the medium supplemented with 50 mM L-Gln; on the medium with 50 mM L-Asn, the HY strain, in contrast to the WT strain, showed the best growth. For high-yielding CPC biosynthesis by the *A. chrysogenum* HY strain, it is desirable to have high concentrations of components in the synthetic medium. In this regard, for further work, we used L-Asn as the sole source of nitrogen at a selected concentration of 50 mM (7.5 g/L).

#### 2.1.2. Determining Doses of PAs for Cultivation of *A. chrysogenum* HY on SN Medium with L-Asn

In the next step, we studied the effect of adding PAs (1,3-DAP and SPD) to the CAA medium on the growth of *A. chrysogenum* WT and *A. chrysogenum* HY colonies (Figure 2). PAs concentrations of 1 mM, 5 mM, and 10 mM were used. It turned out that on the CAA medium, both strains responded rather closely to the addition of both PAs (Figure 2b,c). For both strains, when 1 mM PAs was added, the size of the colonies did not practically differ from the control, and the addition of 5 mM led to a slight increase in their size. The 10 mM dose was toxic to cells, resulting in a significant reduction in colony size. Moreover, the toxic effect for the HY strain was somewhat higher. When 10 mM 1,3-DAP was added, the colony size decreased by ~30% for the WT strain and by ~50% for the HY strain. The addition of 10 mM SPD resulted in a decrease in colony size by ~40% for the WT strain and by ~75% for the HY strain (Figure 2b,c).

As a result, concentrations of 5 mM for 1,3-DAP and 5 mM for SPD were chosen for further work with the HY strain on an SN medium, at which a slight (by 5–10%) growth stimulation was observed. The selection of other synthetic components for SN media (maltose as a source of sugars, KH_2_PO_4_ as a source of phosphorus, trace elements, and so on) was made as a result of long-term optimization of expression and based on previous knowledge of the composition of media for *A. chrysogenum* HY [7].

### 2.2. Submerged Fermentation of A. chrysogenum HY Strain with Exogenous PAs on SN and CP Media

Previously, we optimized the stages of *A. chrysogenum* HY fermentation on CP media with PAs; in particular, we determined the most appropriate time for the introduction of exogenous PAs, which turned out to be the moment of culture transfer from the seed medium to CP medium [11]. In the current study, we performed fermentation of *A. chrysogenum* HY with the same parameters on SN medium (with exogenous PAs) and obtained unexpected results (Figure 3). Up to 72 h of fermentation, both PAs slightly stimulated the production of CPC, which was previously observed on CP media (Figure 3a, [11]). However, from the 96th h of fermentation with SPD, the production of CPC was slowed down compared to the control. By the end of fermentation, the yield of CPC was 14–15% lower than in the control (Figure 3a). At the same time, in samples with 1,3-DAP, an increase in the production of CPC occurred throughout the entire cultivation period on SN medium; by the end of fermentation, the increase in production compared to the control was 12–15%, as in the case of fermentation on CP media (Figure 3a, [11]). Such an opposite effect of 1,3-DAP and SPD on the production of the target SM in filamentous fungi was incomprehensible since earlier studies have observed either a simultaneous stimulation of the production of the target SM [9,10,11] or a decrease in the yield under specially selected conditions [42].

In addition, the decrease in the level of CPC in the samples taken after 96 h, 120 h, and 144 h of fermentation was not accompanied by a decrease in the level of dry biomass compared to the control (Figure 3b). This may indicate that 5 mM SPD is not toxic to *A. chrysogenum* HY in SN medium. This is also consistent with preliminary data obtained on the agarized Czapek-L-Asn medium, where the addition of 5 mM PAs was also non-toxic (Figure 2). A decrease in the yield of CPC against the background of relatively similar biomass led to a decrease in the level of CPC-specific production to ~15% by the end of fermentation compared to the control (Figure 3c). The addition of 1,3-DAP also did not affect the level of dry biomass; as a result, the level of CPC-specific production by the end of fermentation was ~15% higher than in the control and 25–30% higher than with the addition of SPD (Figure 3b,c).

We also performed fermentation on a CP medium for control, where previous results (for CPC yield, dry biomass, and expression of biosynthetic genes from beta-lactam BGCs) were largely replicated [11]. However, we performed additional experiments: qPCR (to determine the gene expression of transport genes from beta-lactam BGCs) and to determine the content of DAC in HPLC samples.

### 2.3. Effect of the Addition of Exogenous PAs to SN and CP Media on Cephems Ratio

To explain the revealed phenomenon of the opposite effect of PAs in SN medium on CPC production, we studied in more detail the HPLC profiles of beta-lactams. Samples taken after fermentation of *A. chrysogenum* HY on SN medium with SPD (after 96 h and later) contained a reduced CPC peak and another major peak, increased compared to the control (Figure 4). This peak corresponded to DAC, a precursor of CPC in the biosynthetic pathway. In 1,3-DAP samples from the SN medium, the CPC peak was increased during all fermentation, compared to the control; the DAC peak remained at the level of other minor by-products as in the fermentation of both PAs on the CP medium [11]. The total percentage level of all other detected minor by-products was approximately the same in the control and samples with both PAs and did not exceed 5–7% of the main cephem fraction (CPC and DAC).

For a more detailed analysis, the total amount of major cephems (CPC and DAC) after fermentation of *A. chrysogenum* HY on SN and CP media was determined (Figure 5). It turned out that the drop in CPC production upon the addition of SPD to SN medium was almost completely compensated by an increase in DAC content (Figure 5a). The total amount of major cephems in SPD samples from SN medium remained at the same level as in 1,3-DAP samples and was 12–15% higher than in the control to the end of fermentation. In general, the trend towards a gradual increase in the total number of major cephems during fermentation with the addition of both SPD and 1,3-DAP compared to the control on SN medium correlates with data for CP media (Figure 5b).

To quantify the disruption at the last stage of CPC biosynthesis in the *A. chrysogenum* HY strain on SN medium with SPD, the percentage of DAC/CPC was determined (Figure 6). After 72 h of fermentation, the ratio of DAC content to CPC in all samples (with additives and in control) on both nutrient media was approximately the same, 12–15% (Figure 6a). However, after 96 h of fermentation on SN medium, when there was a sharp increase in CPC production compared to 72 h (Figure 3a), the addition of SPD resulted in a significant increase in DAC/CPC ratio, up to 30% (Figure 6b). This value continued to rise, and by the end of the fermentation, the DAC/CPC ratio for samples on the SN medium with SPD was ~50% (Figure 6d). At the same time, samples on both media with the addition of 1,3-DAP and samples with SPD on the CP medium had a DAC/CPC ratio close to the controls throughout the entire fermentation period, 10–15% (Figure 6).

The data obtained indicate that on SN medium SPD, but not 1,3-DAP, disrupts the last step of CPC biosynthesis. On the one hand, the total increase in cephems suggests that on an SN medium, SPD still stimulates the biosynthesis of beta-lactams; on the other hand, the decrease in the content of CPC and the accumulation of DAC indicates some additional effect of SPD on the final stage of biosynthesis.

### 2.4. Analysis of the Expression Level of the Biosynthetic and Transport Genes from Beta-Lactam BGCs in A. chrysogenum HY after Fermentation on SN and CP Media with PAs

To explain the phenomenon obtained, which was expressed in the opposite effect of 1,3-DAP and SPD on the SN medium on the yield of CPC in *A. chrysogenum* HY, we studied the level of expression of genes for the biosynthesis and transport of beta-lactams (Figure 7). In particular, we studied the work of genes for all stages of the CPC biosynthesis: (i) pcbAB, which encodes ACV (δ-[L-α-Aminoadipyl]-L-Cysteinyl-D-Valine) synthetase (EC: 6.3.2.26) for the polymerization of L-α-aminoadipic acid, L-cysteine, and L-valine to ACV tripeptide; (ii) pcbC, which encodes isopenicillin N-CoA synthetase (EC: 5.1.1.17) for cyclization of the ACV tripeptide to form isopenicillin N; (iii) cefD1 which encodes isopenicillin N-CoA synthetase (EC: 5.1.1.17) for isopenicillin N activation by acylation for the epimerase reaction; (iv) cefD2 which encodes isopenicillin N-CoA epimerase (EC: 5.1.1.17) for epimerization of isopenicillin N-CoA; (v) CefEF which encodes deacetoxycephalosporin C synthetase (penicillin N expandase, EC: 1.14.20.1)/ deacetoxycephalosporin C hydroxylase (EC:1.14.11.26) for two consecutive reactions of the conversion of penicillin N to deacetoxycephalosporin C (DAOC) and then—DAOC to DAC; (vi) cefG which encodes deacetylcephalosporin-C acetyltransferase (EC: 2.3.1.175) for acetylation of DAC to CPC by acetyl-CoA substrate. We also studied the expression of genes for the transport of intermediate products of CPC biosynthesis: (vii) cefP, which encodes MFS (major facilitator superfamily) transporter for isopenicillin N transport from cytosols to peroxisomes; (viii) cefM which encodes MFS transporter for penicillin N transport from peroxisomes to cytosols; (ix) cefT which encodes MFS transporter for transport of beta-lactam biosynthetic products from the cytoplasm outside the cell. As a control, we also studied the expression of transport genes in samples obtained after fermentation of *A. chrysogenum* HY (with PAs and without additives) on CP media (Appendix A). The expression of biosynthetic genes on CP media was studied previously [11]; the data obtained in the current study on the expression of biosynthetic genes reproduced previous results.

The addition of PAs to the SN medium in *A. chrysogenum* HY generally led to an increase in the expression of beta-lactam biosynthesis genes (by 1.5–4 times) and had no effect on the expression of transport genes (Figure 7). For biosynthetic genes, the upregulation effect of the addition of SPD was close to that of the addition of 1,3-DAP. For biosynthetic genes, the upregulation effect upon the addition of SPD and 1,3-DAP was close. Upregulation of *pcbA*B with the addition of both PAs was observed in the period of 24–120 h of fermentation (Figure 7a). In the early stages, up to 48 h, a two-fold increase in the expression of this gene was observed then it gradually decreased. By the end of fermentation, *pcbAB* upregulation was not detected (dropped to the control level). Upregulation of *pcbC* under the influence of PAs, by 1.2–2 times, was observed after 24 h until the end of fermentation (Figure 7b). After 72 h of fermentation for *pcbC*, a two-fold increase in expression was observed after the addition of 1,3-DAP and a 1.5-fold increase after the addition of SPD. By the end of fermentation, the upregulation caused by the addition of both PAs leveled off and amounted to 1.2–1.4 times. The upregulation of *cefD1* with the addition of PAs was the lowest among all biosynthetic genes, 1.2–1.8 times, in the range of 24–96 h of fermentation, except for 24 h with the addition of 1,3-DAP, where a two-fold increase in gene expression was observed (Figure 7c). At the late stages of fermentation, the expression level of *cefD1* was approximately the same in control and experimental samples. The activation of *cefD2* under the influence of PAs was higher than for *cefD1* and amounted to 2–3 times from 24 h to the end of fermentation (Figure 7d). The addition of 1,3-DAP and SPD led to close *cefD2* upregulation.

Upregulation of “late” genes with the addition of PAs was observed after 72 h of fermentation and until the end of fermentation (Figure 7e,f). The same response of “late” genes to the addition of PAs was previously observed in the CP medium [11]. Upregulation of *cefEF* upon the addition of both PAs to the SN medium was approximately the same, 1.5–2 times (Figure 7e). Upregulation of *cefG* was somewhat stronger; after 72 h, it was 2–3.5 times from the addition of 1,3-DAP and 3–5 times from the addition of SPD (Figure 7f). After the end of fermentation, the upregulation of *cefG* by both PAs was 3–3.5 times, which turned out to be the strongest effect at 144 among all the genes studied on the SN medium (Figure 7). Moreover, the upregulation of *cefG* under the influence of SPD at the late stages of fermentation was the strongest among all observed upregulations. These data are rather unexpected since the addition of SPD along with the upregulation of *cefG* (compared to the control, as well as to 1,3-DAP samples) resulted in a decrease in the CPC yield (Figure 3a) and proportional accumulation of DAC, its metabolic pathway precursor (Figure 4, Figure 5 and Figure 6).

We also studied the expression of the transport genes *cefP*, *cefM*, and *cefT* in response to the introduction of exogenous PAs. However, in our studies on SN medium, we failed to find a significant difference compared to the control for the expression of any of the beta-lactam transporters (Figure 7g–i). The same response was demonstrated by transport genes when PAs were added to the CP medium (Appendix A). Previously, there was also no change in the expression of the *lovT* transport gene from the lovastatin BGC when 1,3-DAP or SPD was added to *Aspergillus terreus* high-yielding strain [42].

Changes in the expression of beta-lactam transporter genes can significantly affect the production of cephalosporin C. It was shown that overexpression of *cefT* in recombinant strains of *A. chrysogenum* HY led to the release of beta-lactam intermediates from the cell, which led to a decrease in the CPC yield [50]. The absence of changes in the expression of genes for the transport of beta-lactam intermediates indicates that the disruption at the last stage of CPC biosynthesis upon the addition of SPD (but not 1,3-DAP) to SN medium does not occur due to a shift in the transport of intermediates.

To determine the effect of the nutrient medium with the addition of exogenous PAs or without additives (control) on the expression of beta-lactam BGCs genes, we studied changes for pairs SN_1,3-DAP_/CP_1,3-DAP_, SN_SPD_/CP_SPD_, and SN_control_/CP_control_ (Figure 8). It turned out that biosynthetic genes are generally downregulated in the SN medium, while transport genes are generally upregulated. Biosynthetic genes were downregulated in SN_control_ compared to the CP_control_ medium; by the end of fermentation, this decrease was 2.2–1.2 times. The addition of 1,3-DAP or SPD to SN and CP media did not significantly change the SN/CP ratio of gene expression, respectively. For example, for *pcbAB*, the expression on the SN medium, both in the control and with the addition of PAs, was reduced to a level of 80–90% of the corresponding values on the CP medium throughout the entire fermentation period (Figure 8a). The expression of *pcbC* on SN medium variants was generally reduced by 60–90% throughout the entire fermentation period, except for 72 h, when the expression level on SN_1,3-DAP_ medium was 1.2 times higher than on CP_1,3-DAP_ (Figure 8b). Comparative expression of *cefD1* revealed an increase in downregulation by the end of fermentation for all variants of the SN medium. By the end of fermentation, downregulation on SN_control_, SN_1,3-DAC_, and SN_SPD_ was more than 50% compared to variants of CP media, respectively. For biosynthetic genes *cefD2*, *cefEF*, and *cefG,* a close ratio for relative expression on SN/CP media without or with PAs was observed, respectively (Figure 8d–f). For these genes at the beginning of fermentation (1 h), there were practically no differences, or expression on the SN variants of the medium was slightly higher (by 5–20%). Then, the level of expression on the SN variants of the medium decreased and amounted to 50–80% of the expression of these genes on the CP variants of the media, respectively.

Taking into account the comparative data on the expression of biosynthetic genes, the data after comparing the expression of the transport genes *cefP*, *cefM*, and *cefT* on two media turned out to be unexpected (Figure 8g–i). Both for the control SN medium and with the addition of PAs, after 24–48 h and until the end of fermentation, upregulation of all transport genes was observed, by 1.2–2.3 times, compared with the corresponding variants on CP media (Figure 7g–i, and Appendix A). This effect was both at the level that occurs when 1,3-DAP is added to the SN medium and in the range that distinguishes gene expression in the SN medium from gene expression in the CP medium (Figure 8). The expression of the key gene *cefG*, which is responsible for the conversion of DAC to CPC, was not changed compared to the addition of 1,3-DAP to the SN medium. Additionally, the expression of the transport gene *CefT*, whose work leads to the export of DAC from the cell, did not change compared with the addition of 1,3-DAP to the SN medium.

As a result, we can conclude that the disruption at the last stage of CPC biosynthesis caused by the addition of spermidine to the SN medium is not associated with disorder in the expression of biosynthetic and transport genes of beta-lactam BGCs.

### 2.5. Effect of the Addition of Exogenous PAs to SN and CP Media on Acetyl-CoA Content

Since it was shown that the observed opposite effect of 1,3-DAP and SPD on the final stage of CPC biosynthesis is not associated with a change in the level of expression of both biosynthetic and transport genes of beta-lactam BGCs, in order to determine another possible reason for this phenomenon, we studied the cellular content acetyl-CoA, which is required for the last step as a substrate for CefG. When PAs were added to the CP medium, the acetyl-CoA content was almost the same compared to the control (Figure 9b). On the SN medium, acetyl-CoA content in the control was about two times lower than on the CP medium (Figure 9a). At the same time, the addition of 1,3-DAP practically did not lead to a change in acetyl-CoA content, while a significant decrease in acetyl-CoA content was observed in samples with added SPD. After 96 h of fermentation with SPD, the decrease in the content of acetyl-CoA was approximately 1.5 times compared to control; after 144 h, it was more than 2 times lower (Figure 9a).

On the SN medium, which is poorer in nutrient resources than the CP medium, when SPD is added, homeostasis of the acetyl-CoA content is no longer maintained; the acetyl-CoA pool is depleted. Possibly 5 mM SPD, but not 1,3-DAP, specifically depletes the acetyl-CoA pool (required for the conversion of DAC to CPC) on SN medium. This may be due to the significantly higher activity of SAAT, the acetyl-CoA-consuming enzyme, for SPD catabolism [51]. The data obtained on the SN medium, expressed in the depleting effect of SPD on the acetyl-CoA pool, suggest that this effect can be transferred to the efficiency of the last stage of CPC biosynthesis (Figure 5a and Figure 6).

## 3. Discussion

Comparative fermentation of the two strains showed that the *A. chrysogenum* HY strain during multi-round mutagenesis did not undergo changes that reduced the consumption of the studied five nitrogen sources, such as NaNO_3_, NH_4_Cl, urea, L-asparagine, and L-glutamine (Figure 1b,c). Such phenotypic behavior also indirectly indicates the absence of disruption in the *AcareA* and *AcareB* genes, regulating N consumption in *A. chrysogenum*.

On the CP medium, the total yield of major cephems (CPC and DAC) was 2–2.3 times higher both for the control and with the addition of SPD or 1,3-DAP (Figure 5). This indicates the balance of this medium in terms of nutritional composition for the biosynthesis of CPC with PAs by the *A. chrysogenum* HY. It is likely that this medium masks undesirable effects from possible crosstalk of CPC biosynthesis and PAs metabolism. However, these effects appeared on SN medium with SPD in the late fermentation period, when nutrient resources are limited, where specific changes were observed in comparison with control and 1,3-DAP, expressed in the accumulation of the CPC precursor, in proportion to the decrease in the content of CPC. On SN medium without additives, the ratio of cephems (DAC/CPC) was very close to that on CP medium and amounted to 10–15% (Figure 6). This indicates the effectiveness of this medium for studying the mechanisms of CPC biosynthesis but not its high-yielding production.

The absence of upregulation of transport genes (*cefP*, *cefM*, and *cefT*) against the background of upregulation of biosynthetic genes with the addition of PAs does not prevent an increase in the final production of cephems (Figure 5 and Figure 7). In particular, the expression of *cefT* was practically unchanged by the addition of PAs during fermentation on both media (Figure 7i, Appendix A). However, an increase in the release of major cephems, DAC, and CPC, into the culture liquid was observed (Figure 5). Perhaps this beta-lactam transporter has some “safety margin” for its functioning, or cephems may be released from *A. chrysogenum* cells in some other way, as discussed earlier [50]. On the other hand, the lack of modification in the expression of the *cefM* transporter and upregulation in the *CefD2* expression should expect an accumulation of penicillin N inside peroxisomes. Such an effect of PAs on the expression of these genes is observed both in the SN medium (Figure 7d,h) and in the CP medium ([11], Appendix A). In our work, in accordance with the methodology, the profile of the beta-lactam fraction was determined from the culture liquid, and the content of beta-lactams in peroxisomes was not measured separately. Therefore, we could only measure penicillin N, which is secreted into the culture liquid, and its amount was not increased; it was at the level of minor beta-lactam impurities (Figure 4). However, in order to clearly trace the dynamics of changes, we studied the production of beta-lactams and the level of gene expression every day, starting from 24 h of fermentation; we also measured the level of gene expression an hour after the start of fermentation, when the production of beta lactams was relatively low (Figure 3 and Figure 7, Appendix A). Such consistent observation of the dynamics of beta-lactam production and expression of genes from beta-lactam BGCs provide an opportunity to assess the efficiency of beta-lactam biosynthesis in general. Since we observe an increase in the content of DAC and CPC, the compounds from the next stages of the beta-lactam biosynthetic pathway, with the addition of PAs, it can be assumed that there is no accumulation of the intermediate penicillin N in peroxisomes. This is not direct but quite significant indirect evidence. Perhaps the CefM transporter, like CefT, has a certain “margin of safety” for its functioning, which makes it possible to effectively ensure the transport flow of an increased pool of beta-lactams without changing the level of expression of the corresponding gene. The same phenomenon was described for another improved fungal strain *Aspergillus terreus*, a high-yielding producer of lovastatin. The addition of PAs led to an increase in the production of lovastatin (by 25–45%) and upregulation of biosynthetic *lov*-genes against the background of no changes in the expression of the transport gene *lovT* [42].

The yield of CPC during the fermentation of *A. chrysogenum* strongly depends on the efficiency of the final stage of biosynthesis, which is associated with DAC acetylation by the catalytic activity of CefG [43]. In this case, it is important to have a sufficient amount of both the donor for acetylation, acetyl-CoA and CefG itself. Acetyl-CoA is a central metabolite in carbon and energy metabolism [52]. For example, a decrease in the intracellular ATP content in recombinant *A. chrysogenum* strains (with overexpression of the plasma membrane H^+^-ATPase) leads to the accumulation of DAC and a decrease in CPC production since ATP is required for the synthesis of cytoplasmic acetyl-CoA [40]. The addition of PAs during the fermentation of *A. chrysogenum* HY on a CP medium leads to an upregulation of *cefG* by 6–8 times [11]. This is one of the most important molecular bases explaining the increase in CPC production upon the addition of PAs to the CP medium [11]. In a current study on SN medium, *cefG* upregulation (as for other CPC biosynthesis genes) was slightly lower at 3–4 fold (Figure 7f and Figure 8f). However, this did not influence the increase in CPC production during the cultivation of *A. chrysogenum* HY on SN medium with 1,3-DAP; like on a CP medium, it was 12–15% compared to control (Figure 4). At the same time, the addition of spermidine to SN medium decreased the yield of CPC by 14–15% compared to control (Figure 3a and Figure 5). However, simultaneously with the decrease in the amount of CPC, a significant increase in the content of DAC was observed (Figure 4a). The percentage of DAC/CPC at the end of fermentation with spermidine was ∼5 times higher than with fermentation with 1,3-DAP or control (Figure 6). As a result, the ratio in the fraction of major cephems (CPC and DAC) changed, but their total amount turned out to be very close to the total cephems obtained during fermentation with 1,3-DAP (Figure 4a). This may indicate insufficient efficiency of the reaction of the final stage of CPC biosynthesis. At the same time, the previous biosynthetic steps leading to the formation of DAC function very closely when both SPD or 1,3-DAP are added because they provide the same total amount of “late” products of beta-lactam biosynthesis (CPC and DAC) (Figure 10). However, then about a third of the cephems were not metabolized into CPC during fermentation with SPD, while only about one-tenth of the cephems remained in the DAC fraction during fermentation with DAP or in the control (Figure 3a and Figure 5). At the same time, the level of expression of both biosynthetic and transport genes upon the addition of spermidine and DAP was quite close (Figure 7). This may indicate that the disruption in the last step of CPC biosynthesis is associated with a lack of acetyl-CoA during the fermentation of *A. chrysogenum* HY on SN medium supplemented with SPD.

For the catabolism of higher polyamines, such as spermidine or spermine, N^1^-acetylation is used in the first stage [30]. Acetylated polyamines are then available for transport out of the cell or catabolism by amine oxidases [30,31,32]. The N^1^-acetylation of PAs is catalyzed by the SSAT enzyme, which has not yet been described for *A. chrysogenum*; therefore, we cannot currently test our hypothesis on crosstalk of the pathways of CPC biosynthesis and SPD catabolism at the level of the common substrate, acetyl-CoA. However, it is known that SSAT is universally found in all living organisms and plays an important role in the regulation of polyamine homeostasis [30]. With an increase in the content of PAs, the upregulation of the *ssat* gene occurs, which leads to a decrease in their number as a result of triggering catabolic reactions. It has also been shown that 1,3-DAP acetylation is 1.5–2 orders of magnitude less efficient than spermidine acetylation by this enzyme [51]. This explains the opposite effect of 1,3-DAP and SPD on CPC production in SN medium. SPD depletes the acetyl-CoA pool in the resource-limited SN medium, leading to DAC accumulation, and 1,3-DAP practically does not consume acetyl-CoA since, for it, the N^1^-acetylation reaction is less efficient than for DAC acetylation.

To test this hypothesis, we examined the content of acetyl-CoA in both media (Figure 9). It turned out that in the control samples (without the addition of polyamines), the amount of acetyl-CoA in the SN medium was approximately two times lower than in the CP medium. This may be due to the composition of the medium, which strongly affects the acetyl-CoA content in fungi [53]. At the same time, the addition of polyamines to the CP medium did not affect the content of acetyl-CoA, while the addition of SPD, but not 1,3-DAP, to the SN medium led to a significant decrease in acetyl-CoA content. By the end of fermentation (144 h), this decrease was more than two times compared to the control (Figure 9b). The obtained data indirectly confirm our hypothesis about the intersection of the pathways of beta-lactam biosynthesis and polyamine catabolism at the level of the common substrate, acetyl-CoA (Figure 10).

The obtained data on the opposite effect of polyamines at the molecular level are important both from a fundamental point of view since they indicate the intersection of the biosynthesis of cephalosporin C and the metabolism of polyamines and for practical use in order to select optimal conditions for increased production.

## 4. Materials and Methods

### 4.1. Materials

1,3-diaminopropane (1,3-DAP) and spermidine (Spd) were obtained from MP Biomedicals, Santa Ana, CA, USA.

### 4.2. Strains of Microorganisms

*A. chrysogenum* ATCC 11550 (WT, wild-type Brotzu isolate, [54]) and *A. chrysogenum* RNCM 408D (HY, high-yielding CPC producer, derived from the WT, [7]) were used in this work.

### 4.3. Cultivation of A. chrysogenum Strains on Agarized Media with PAs

*A. chrysogenum* strains were cultivated on (i) agarized complex (CPA) medium (40 g/L maltose, 10 g/L peptone, 20 g/L malt extract, 25 g/L agar, pH 7.0–7.4); (ii) or agarized Czapek-Dox (CDA) medium (30 g/L sucrose, 2 g/L NaNO_3_, 1 g/L K_2_HPO_4_, 0.5 g/L MgSO_4_ × 7H_2_O, 0.5 g/L KCl, 0.01 g/L FeSO_4_ × 7 H_2_O, 25 g/L agar, pH 7.0–7.4); (iii) or agarized Czapek-N medium (30 g/L sucrose, 1 g/L K_2_HPO_4_, 0.5 g/L MgSO_4_ × 7H_2_O, 0.5 g/L KCl, 0.01 g/L FeSO_4_ × 7 H_2_O, 25 g/L agar, pH 7.0–7.4), supplemented with NaNO_3_, NH_4_Cl, urea, L-Asp, or L-Glu in the concentration range 1–50 mM; (iv) or agarized LPE medium (10 g/L glucose, 20 g/L yeast extract, 15 g/L NaCl, 10 g/L CaCl_2_, 25 g/L agar, pH 6.8). Agarized Czapek-L-Asn (CAA) medium was supplemented with 1,3-DAP, or SPD, in the concentration range of 1–10 mM or used without additions (control).

### 4.4. Light Microscopy

Light microscopy in native conditions was performed using a Carl Zeiss Jena microscope (Carl Zeiss, Germany) at ×1000 magnification.

### 4.5. Submerged Fermentation of A. chrysogenum HY Strain with Exogenous PAs

*A. chrysogenum* HY strain was routinely cultured on CPA slants. To prepare the strain for antibiotic production, it was inoculated from CPA on LPE slants, incubated 10 days at 28 °C, the whole content collected from agar with 5 mL 0.9% NaCl, transferred to 25 mL of the defined (DP) medium (28 g/L yeast extract, 28 g/L malt-extract, 10 g/L peptone, 4 g/L chalk, 20 g/L soybean oil, pH 7.2) in 250 mL Erlenmeyer flasks, and incubated on a rotary shaker at 220–240 rpm at 28 °C. After 48 h of growth, the mycelium was separated from the culture liquid by filtration and collected on a Flag Grid filter with a pore diameter of 15–20 µm (EuroFlag, Moscow, Russia), washed with 10 volumes of 100 mM potassium phosphate buffer (PPB, pH 6.4). In total, 4 g of the mycelium wet biomass was inoculated into 55 mL of the synthetic (SN) medium (60 g/L maltose, 7.5 g/L L-asparagine, 2 g/L KH_2_PO4, 5 g/L CaCO_3_, supplemented with microelements: 18 mg/L CuSO_4_×5H_2_O, 150 mg/L ZnSO_4_ × 7H_2_O, 30 mg/L MnSO_4_ × 7H_2_O, 70 mg/L FeSO_4_ × 7H_2_O, pH 6.2–6.4) or into 55 mL of complex (CP) medium (105 g/L corn extract, 60 g/L corn dextrin, 20 g/L corn starch, 3 g/L KH_2_PO_4_, 5 g/L glucose, 3.5 g/L MgSO_4_, 14 g/L (NH_4_)_2_SO_4_, 11 g/L chalk, 20 g/L soybean oil; supplemented with microelements: 18 mg/L CuSO_4_ × 5H_2_O, 150 mg/L ZnSO_4_ × 7H_2_O, 30 mg/L MnSO_4_ × 7H_2_O, 70 mg/L FeSO_4_ × 7H_2_O, pH 6.2–6.4). Fermentation was performed in 750-mL Erlenmeyer flasks for 144 h (240 rpm) at 28 °C for the first 24 h and at 24°C for the rest of the process. SN and CP media were supplemented with 5 mM 1,3-DAP or SPD, or no additions were made (control).

### 4.6. Determination of Dry Biomass

Aliquots (2 mL), which included medium and cells, were taken after 24 h, 48 h, 72 h, 96 h, 120 h, and 144 h of growth, centrifuged in a 15 mL falcon at 4800× *g*, 10 min, washed three times with 10 volumes of H_2_O and placed in a thermostat at 80 °C. Drying was carried out for 48–72 h until a constant weight was established. Dry biomass was determined by the difference between the weight of dried cells and empty falcon. Data represent triplicates from four separate experiments, with the mean and SEM displayed.

### 4.7. HPLC Analysis of Beta-Lactams

To determine the yield of CPC, DAC, and minor by-products of the biosynthesis of beta-lactams, aliquots (0.5 mL) of the culture broth were taken after 24 h, 48 h, 72 h, 96 h, 120 h, and 144 h of growth (under the conditions, as described in Section 4.4.), centrifuged in a 1.5 mL Eppendorf at 16,000× *g*, 10 min. The concentration of beta-lactams in the supernatant was determined on the Agilent 1200 liquid chromatograph (Agilent Technologies Inc., Santa Clara, CA, USA) from 10 μL of the samples. The chromatographic column YMC-Pack ODS-A (YMC Co., Tokyo, Japan) with particle size 5 µm was used. Chromatography conditions: a mobile phase consisting of 40% tetrabutylammonium hydroxide: acetonitrile adjusted to pH 7.0 ± 0.1 with 10% phosphoric acid (72.5:27.5, *v*/*v*) at flow rate 1.0 mL/min and UV detection at 254 nm. Data represent triplicates from four separate experiments, with the mean and SEM displayed.

### 4.8. Determination of Cellular Acetyl-CoA

Aliquots (0.5 mL) of the culture broth were taken at 96 h and 144 h of growth, washed two times with 10 volumes of H_2_O, lyophilized, and stored at −80 °C. Acetyl-CoA was determined as described previously [55] with some modifications. Boiling ethanol (1 mL) was added to lyophilized cells, carefully treated with glass beads (D = 500 μm) for 10 min on a vortex, centrifuged at 16,000× *g*, 10 min. The supernatant was vacuum dried and re-suspended in 200 μL ddH_2_O. The acetyl-CoA solution thus obtained was analyzed using an acetyl-CoA assay kit (Merck, Kenilworth, NJ, USA). The resulting concentration of acetyl-CoA was normalized by dry cell weight. Data represent triplicates from three separate experiments, with the mean and SEM displayed.

### 4.9. Preparation of Total RNA and cDNA Synthesis and qPCR Analysis

Cell samples for total RNA extraction were taken at 1 h, 24 h, 48 h, 72 h, 96 h, 120 h, and 144 h of growth, filtered, washed with PBS, lyophilized, and stored at −80 °C. The total RNA preparation and cDNA synthesis were carried out as described previously [36,50]. qPCR reactions were performed with previously developed primer pairs for analysis of the expression of genes for the biosynthesis and transport of beta-lactams (*pcbAB*, *pcbC*, *cefD1*, *cefD2*, *cefEF*, *cefG*, *cefP*, *cefM*, and *cefT*) (Table 1) [36,40,50]. Reactions and processing of the results were carried out in accordance with the protocol [36]. To normalize the data of expression levels, we used a previously designed pair of primers for the housekeeping γ-actin gene [50]. Data represent triplicates from three separate experiments, with the mean and SEM displayed.

### 4.10. Statistical Analysis

The experimental data were expressed as mean value ± standard error of mean (SEM) calculated from three parallel experiments. The statistical analysis was performed by one-way analysis of variance (ANOVA) using Microsoft Excel. Differences described by *p* ≤ 0.05 were considered significant.

## 5. Conclusions

We have shown that exogenous administration of spermidine, but not 1,3-DAP, leads to a change in the ratio of cephems on a specially selected synthetic medium, where the compensating effect of other environmental factors is removed. The total content of cephems increases, but the amount of cephalosporin C decreases. One of the possible explanations is associated with the depletion of the pool of the common substrate, acetyl-CoA, which is necessary both for the biosynthesis of cephalosporin C and for the catabolism of spermidine, but not 1,3-DAP. This hypothesis is confirmed by the fact that the addition of spermidine, but not 1,3-DAP, to the synthetic medium led to a significant, more than two-fold, decrease in the cellular content of acetyl-CoA compared to the control. This phenomenon is important both for understanding the molecular basis of the effect of polyamines on high-yielding fungal strains and for the targeted creation of strains and the selection of nutrient media for increased production of target secondary metabolites.

## Figures and Tables

**Figure 1 ijms-23-14625-f001:**
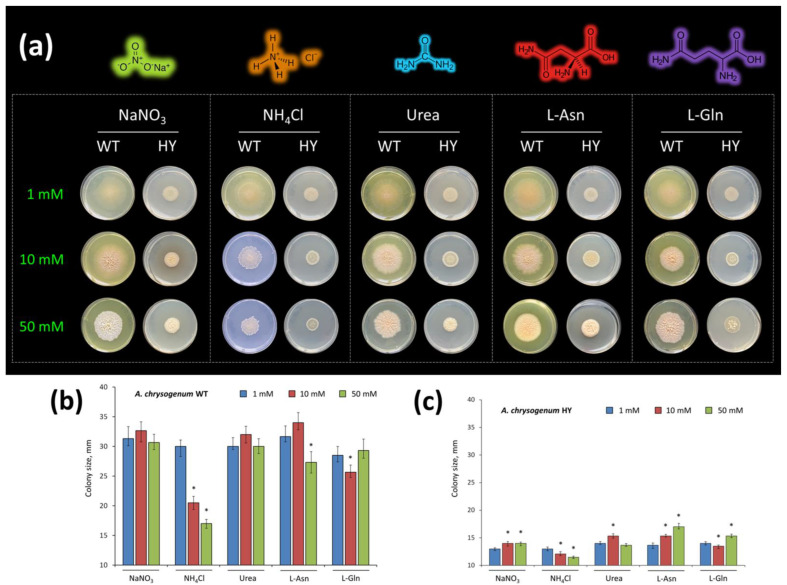
Growth of *A. chrysogenum* wild-type (WT) and high-yielding (HY) strains on agarized Czapek-N medium, supplemented with NaNO_3_, NH_4_Cl, or urea, or L-asparagine, or L-glutamine as sole nitrogen source at the concentration of 1 mM, 10 mM, or 50 mM after incubation for 20 days at 26 °C. (**a**) Phenotype on different media; (**b**) colony diameter size of *A. chrysogenum* WT strain, mm; (**c**) colony diameter size of *A. chrysogenum* HY strain, mm. Statistical significance, * *p* ≤ 0.05, as compared with the control (strain, cultivated on medium with 1 mM additions).

**Figure 2 ijms-23-14625-f002:**
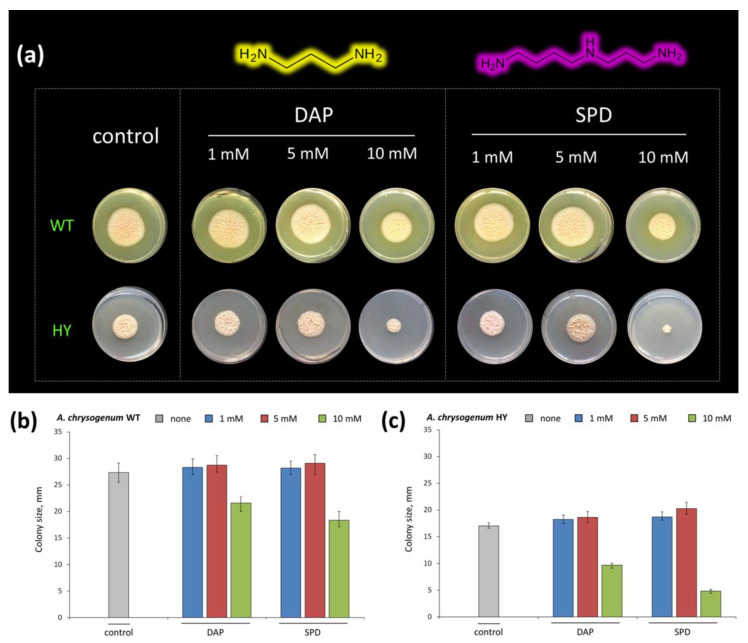
Growth of *A. chrysogenum* wild-type (WT) and high-yielding (HY) strains on agarized Czapek–L-Asn medium, supplemented with 1,3-diaminopropane (DAP) or spermidine (SPD)—both at the concentration of 1 mM, 5 mM, or 10 mM, or without additions (control). Cultivation for 20 days at 26 °C. (**a**) Phenotype on different media; (**b**) colony diameter size of *A. chrysogenum* WT strain, mm; (**c**) colony diameter size of *A. chrysogenum* HY strain, mm.

**Figure 3 ijms-23-14625-f003:**
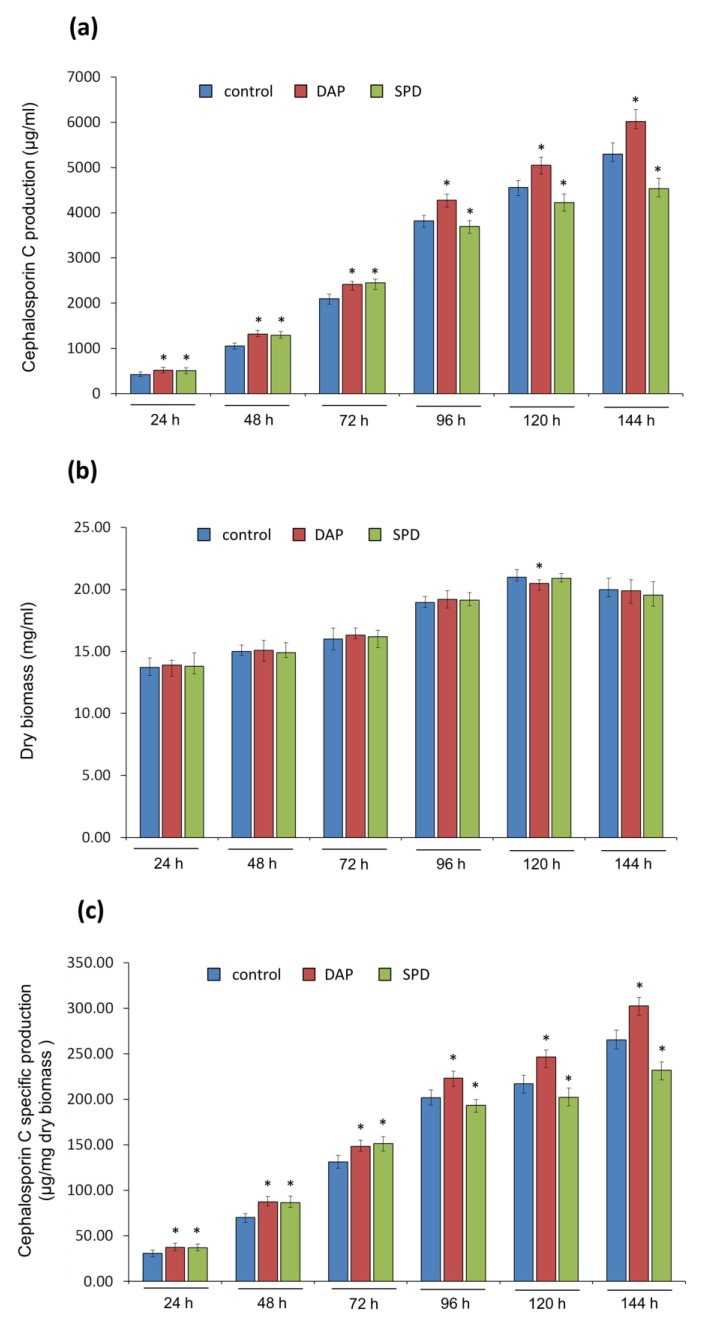
Effect of the addition of 5 mM 1,3-diaminopropane (DAP) or 5 mM spermidine (SPD) on cephalosporin C (CPC) production and growth of *A. chrysogenum* HY strain during fermentation on synthetic (SN) medium: (**a**) CPC production; (**b**) dry weight; (**c**) CPC specific production. Samples were taken at 24 h, 48 h, 72 h, 96 h, 120 h, and 144 h. Data are means ± SD, *n* = 3. Statistical significance, * *p* ≤ 0.05, as compared with the control (strain, cultivated on medium without PAs additions).

**Figure 4 ijms-23-14625-f004:**
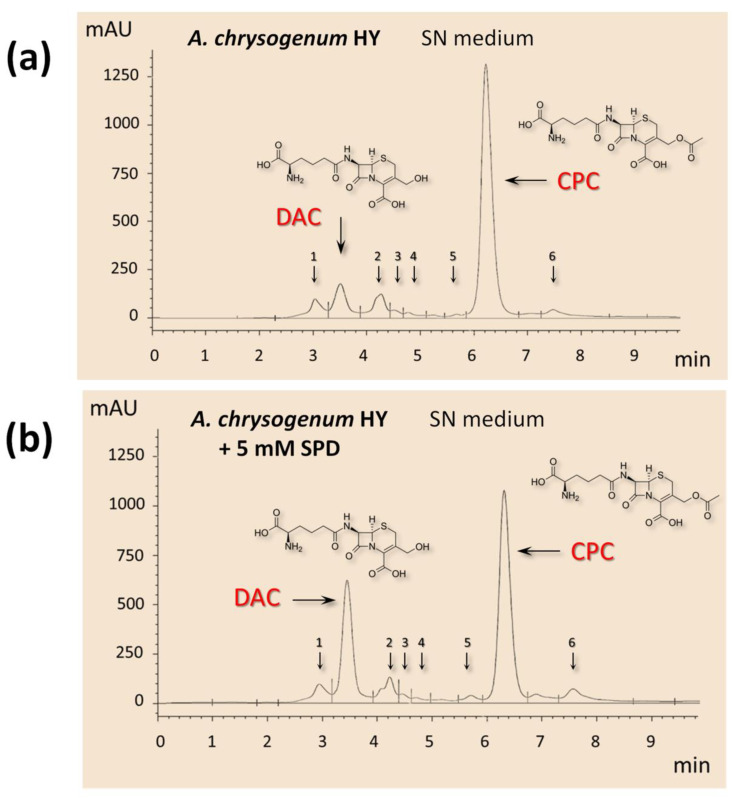
HPLC analysis of beta-lactam production of the *A. chrysogenum* high-yielding (HY) strain after 144 h of cultivation on synthetic (SN) medium: (**a**) without additives (control); (**b**) with 5 mM spermidine (SPD), added at the starting point of cultivation. Arrows show chromatographic peaks corresponding to deacetylcephalosporin C (DAC), cephalosporin C (CPC), and minor by-products.

**Figure 5 ijms-23-14625-f005:**
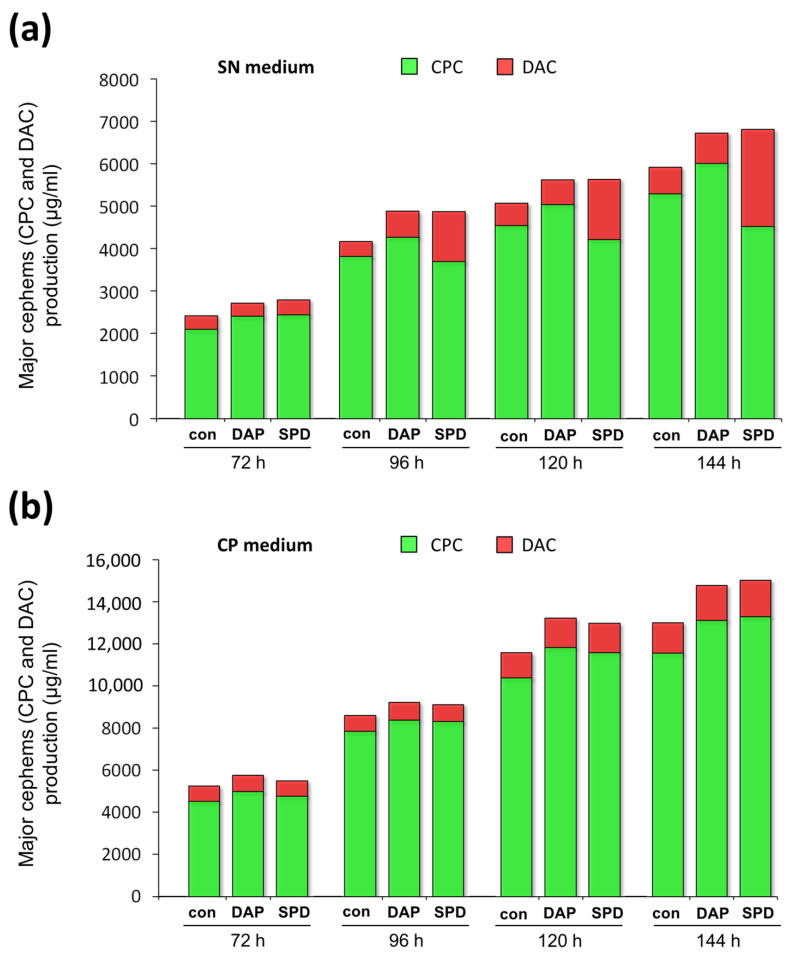
Effect of addition of 5 mM 1,3-diaminopropane (DAP) or 5 mM spermidine (SPD) on the production of cephems (DAC-deacetylcephalosporin C and CPC-cephalosporin C) by *A. chrysogenum* HY strain. Cultivation on: (**a**) synthetic (SN) medium; (**b**) complex (CP) medium. Samples were taken at 72 h, 96 h, 120 h, and 144 h. Data are means ± SD, *n* = 3.

**Figure 6 ijms-23-14625-f006:**
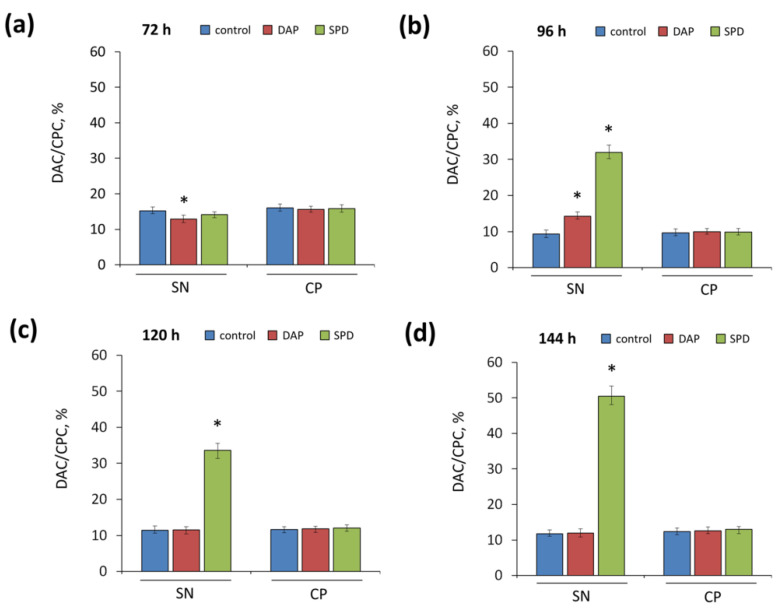
Relative DAC/CPC production (%) after cultivation of *A. chrysogenum* HY strain on synthetic (SN) and complex (CP) media with 5 mM 1,3-diaminopropane (DAP), or 5 mM spermidine (SPD), or without additions (control) for (**a**) 72 h; (**b**) 96 h; (**c**) 120 h; (**d**) 144 h. Data are means ± SD, *n* = 3. Statistical significance, * *p* ≤ 0.05, as compared with the control (strain, cultivated on medium without PAs additions).

**Figure 7 ijms-23-14625-f007:**
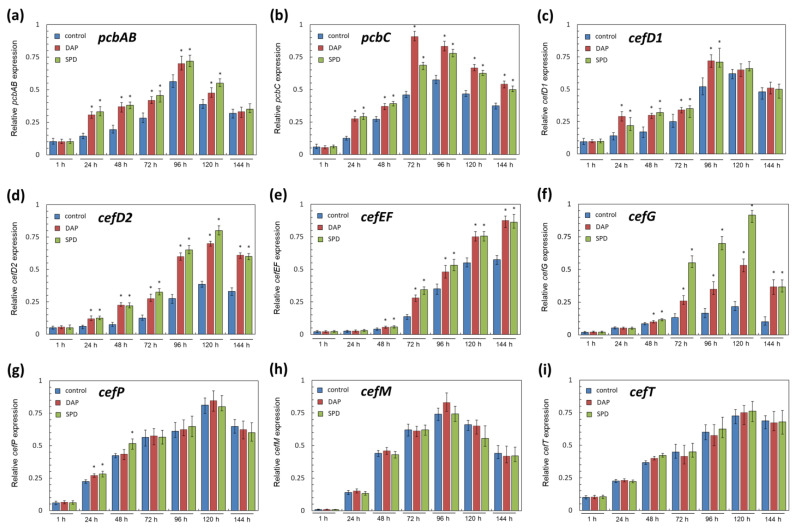
Expression dynamics of (**a**) *pcbAB*; (**b**) *pcbC*; (**c**) *cefD1*; (**d**) *cefD2*; (**e**) *cefEF*; (**f**) *cefG*; (**g**) *cefP*; (**h**) *cefM*; (**i**) *cefT* genes in *A. chrysogenum* HY strain after the addition 5 mM 1,3-diaminopropane (DAP), or 5 mM spermidine (SPD). After 1, 24, 48, 72, 96, 120, and 144 h of fermentation on synthetic (SN) medium. Data are means ± SD, *n* = 3. Statistical significance, * *p* ≤ 0.05, as compared with the control (strain, cultivated on medium without PAs additions).

**Figure 8 ijms-23-14625-f008:**
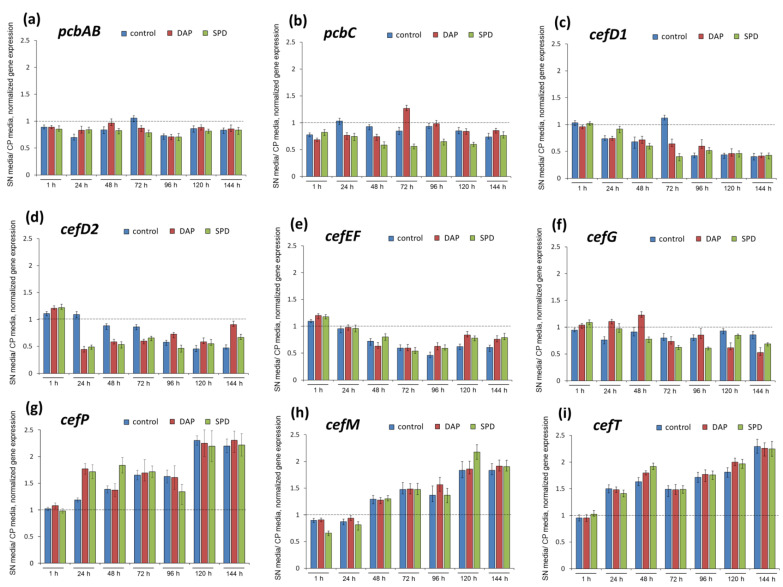
Relative expression on synthetic (SN)/ complex (CP) media for *cef* genes in *A. chrysogenum* high-yielding (HY) strain: (**a**) *pcbAB*; (**b**) *pcbC*; (**c**) *cefD1*; (**d**) *cefD2*; (**e**) *cefEF*; (**f**) *cefG*; (**g**) *cefP*; (**h**) *cefM*; (**i**) *cefT*. Samples were taken at 1 h, 24 h, 48 h, 72 h, 96 h, 120 h, and 144 h of cultivation *A. chrysogenum* HY on SN medium or CP medium, both supplemented with 5 mM 1,3-diaminopropane (DAP), or 5 mM spermidine (SPD), or without addition (control). The dashed lines show a comparative level of gene expression in the *A. chrysogenum* HY strain cultivated on CP medium. Data are means ± SD, *n* = 3.

**Figure 9 ijms-23-14625-f009:**
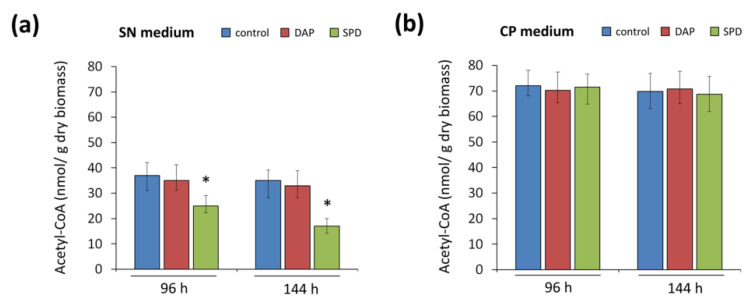
Effect of addition of 5 mM 1,3-diaminopropane (DAP) or 5 mM spermidine (SPD) on the cellular content of acetyl-CoA in *A. chrysogenum* strain HY. Cultivation on: (**a**) synthetic (SN) medium; (**b**) complex (CP) medium. Samples were taken at 96 h and 144 h. Data are means ± SD, *n* = 3. Statistical significance, * *p* ≤ 0.05, as compared with the control (strain, cultivated on medium without PAs additions).

**Figure 10 ijms-23-14625-f010:**
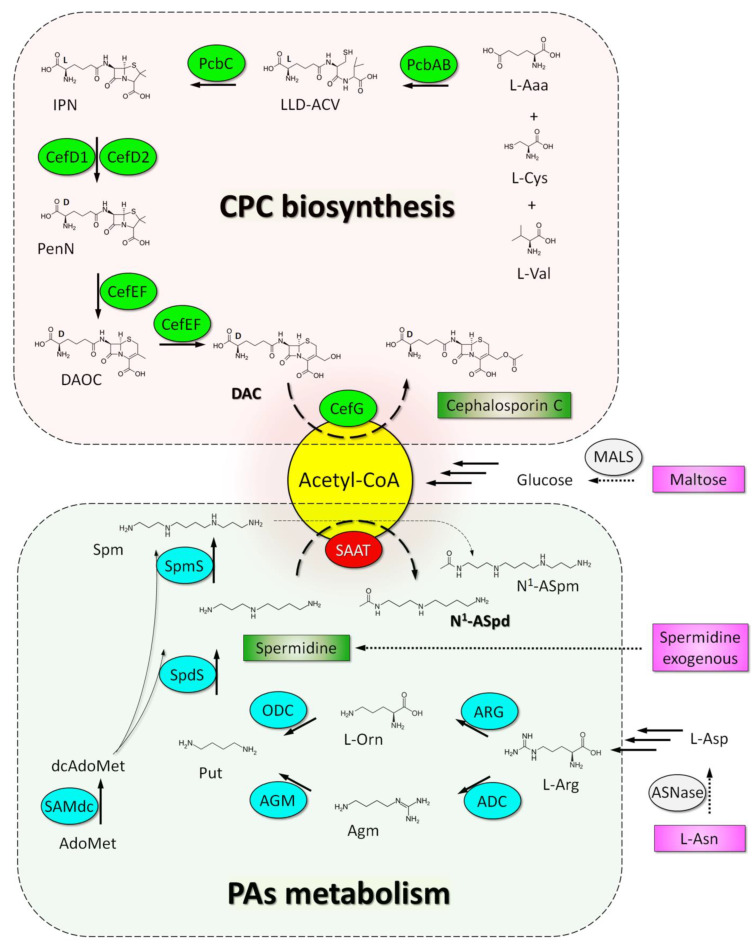
Proposed mechanism for intersection of cephalosporin C (CPC) biosynthesis and polyamine (PAs) catabolism at the level acetyl-coenzyme A (Acetyl-CoA) consumption during cultivation of *A. chrysogenum* high-yielding (HY) strain on synthetic (SN) medium, supplemented with spermidine. Enzymes for the biosynthesis of CPC filled in green: PcbAB—ACV (δ-[L-α-Aminoadipyl]-L-Cysteinyl-D-Valine) synthetase (EC: 6.3.2.26); PcbC—isopenicillin N-synthase (EC: 1.21.3.1); cefD1—isopenicillin N-CoA synthetase (EC: 5.1.1.17); cefD2—isopenicillin N-CoA epimerase (EC: 5.1.1.17); CefEF—deacetoxycephalosporin C synthetase (penicillin N expandase, EC: 1.14.20.1)/deacetoxycephalosporin C hydroxylase (EC:1.14.11.26); CefG—deacetylcephalosporin-C acetyltransferase (EC: 2.3.1.175). Enzymes for biosynthesis of PAs filled in blue: ARG—arginase (L-arginine aminohydrolase, EC 3.5.3.1); ADC—arginine decarboxylase (EC: 4.1.1.19); ODC –ornithine decarboxylase (EC: 4.1.1.17); AGM—agmatinase (EC: 3.5.3.11); SAMdc—S-adenosylmethionine decarboxylase (EC: 4.1.1.50); SpdS—spermidine synthase (EC: 2.5.1.16); SpmS—spermine synthase (EC: 2.5.1.22). Enzyme for catabolism of PAs filled in red: SSAT—spermidine/spermine-N^1^-acetyltransferase (EC: 2.3.1.57). Other enzymes filled in gray: MALS—maltase (α-1,4-glucosidase, EC 3.2.1.20); ASNase—L-asparaginase (EC 3.5.1.1). Compounds of the CPC biosynthetic pathway: L-Aaa—L-α-aminoadipic acid; LLD-ACV—δ-(L-α-aminoadipoyl)-L-cysteinyl-D-valine; IPN—isopenicillin N; PenN—penicillin N; DAOC—deacetoxycephalosporin C; DAC—deacetylcephalosporin C. PAs metabolic pathway compounds: L-Orn—L-ornithine; Agm—agmatine (4-aminobutyl-guanidine); Put—putrescine; AdoMet—S-adenosyl-L-methionine; dcAdoMet—decarboxylated S-adenosyl-L-methionine; Spm—spermine; N^1^-ASpd—N^1^-acetylspermidine; N^1^-ASpm—N^1^-acetylspermine. Exogenous compounds are filled in pink.

**Table 1 ijms-23-14625-t001:** Primers used for RT-PCR analysis.

Primer	Gene	Product, Function	Oligonucleotide (Sequence 5 → 3)	GenBank, Source
actq1	*act1*	γ-Actin, a major component of the cytoskeleton	CCGGTTTCGCCGGTGATGATGCT	JN836733.1, [50]
actq2			TGCTCAATGGGGTAGCGCAG	
pcbABq3	*pcbAB*	δ-(L-α-Aminoadipyl)-L-Cysteinyl-D-Valine synthetase	AGGCATCGTCAGGTTGGCCG	E05192.1, [36]
pcbABq4			CCGGAGGGGCCATACCACAT	
pcbCq1	*pcbC*	Isopenicillin N-synthase	CTAGGTCGCGACGAGGACTTCT	M33522.1, [36]
pcbCq2			CACGTCGGACTGGTACAACACC	
cefD1q1	*cefD1*	Isopenicillin N-CoA synthetase	CCCCGGTGAGGAAGATGCGT	AJ507632.2, [36]
cefD1q2			TCGATCTCCGCCTTGGACGC	
cefD2q1	*cefD2*	Isopenicillin N-CoA epimerase	ACAGGATGGAGAGGAGCACCTTG	
cefD2q2			TCGTAGAGCTCGCGGGGCTA	
cefEFq3	*cefEF*	Deacetoxycephalosporin C synthetase/ hydroxylase	GTCGAGTGCGATCCCCTCCT	AJ404737.1, [40]
cefEFq4			CGAATTCTCCGTCCACCTCG	
cefGq3	*cefG*	Deacetylcephalosporin-C acetyltransferase	CTCCTGGAGCCATATGGAAGCGC	M91649.1, [40]
cefGq4			GGTGCGCAGCTTGGTTCGAGAC	
cefPq1	*cefP*	Multidrug efflux pump protein, putative role in isopenicillin N transport	AATGCGACCCCGAGGAGTACGT	AM231816.1, [36]
cefPq2			CCATCCCAGGAATGTTGTCGGC	
cefMq1	*cefM*	Multidrug efflux pump protein, putative role in penicillin N transport	TTTATCCAGGAGGAGCGCGGTC	AM231815.1, [36]
cefMq2			TGTCGTAGGCGGTTCACCTTGC	
cefTq3	*cefT*	Multidrug efflux pump protein	TGTTGTCGGATTCGGTGTCGG	AJ487683.1, [50]
cefTq4			TTCCACATATCGGCAAGGGTGC	

## Data Availability

The data presented in this study are contained within the article.

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
