# Peer review of "Spermidine and 1,3-Diaminopropane Have Opposite Effects on the Final Stage of Cephalosporin C Biosynthesis in High-Yielding Acremonium chrysogenum Strain"

_ijms, 2022, doi:10.3390/ijms232314625_

Round 1

Reviewer 1 Report

The article entitled "Spermidine and 1,3-Diaminopropane have opposite effects on the Final Stage of Cephalosporin C Biosynthesis in High-Yielding Acremonium chrysogenum Strain" describes that the addition of 1,3-diaminopropane gives rise to an increase in CPC production, unlike the addition of spermidine, the latter showing an opposite effect and accumulation of DAC. This is attibuted to the fact that spermidine depletes the pool of acetyl coenzyme A on synthetic media, thus leading to accumulation of DAC.

The article is well written and easy to read, although there are some points that should be addressed and clarified.

Major points:

Why did not the authors check acetyl-CoA levels after the addition of polyamines to confirm their hypothesis?

Due to the upregulation in the CefD2 expression and the lack of modification in the expression of the cefM transporter, one should expect accumulation of penicillin N inside microbodies. Did authors check this?

What is the purpose of Supplementary Figure S2? Authors say that it refers to the expression dynamics on synthetic (CP) medium. Is CP medium synthetic? Do authorms mean on SN medium? If so, results should be similar to those shown in Figure 7, but some differences can be seen between both figures. Please, clarify this issue.

Minor points:

Line 108. Why authors say "A. chrysogenum HY attenuated strain"? Information about "attenuation" is missing in the text

A. chrysogenum should be written in italics throughout the text

Some cyrillic characters appear in several parts of the text (e.g. line 358). Please, correct it.

Line 431. SN1,3-DAC/CP1,3-DAC. I think authors mean to say SN1,3-DAP/CP1,3-DAP. Please, correct it. 

Author Response

Responses to Reviewer #1

Comments and Suggestions for Authors

Question: The article entitled "Spermidine and 1,3-Diaminopropane have opposite effects on the Final Stage of Cephalosporin C Biosynthesis in High-Yielding Acremonium chrysogenum Strain" describes that the addition of 1,3-diaminopropane gives rise to an increase in CPC production, unlike the addition of spermidine, the latter showing an opposite effect and accumulation of DAC. This is attibuted to the fact that spermidine depletes the pool of acetyl coenzyme A on synthetic media, thus leading to accumulation of DAC.

The article is well written and easy to read, although there are some points that should be addressed and clarified.

Response: We thank the reviewer for the detailed and critical analysis of the manuscript, valuable comments and advices. We have responded to all questions and made adjustments and corrections to the indicated places in accordance with your recommendations. Recommended changes, amendments have been made and are listed in detail in sections below. In the revised manuscript, we marked the changes with yellow font and the point by point responses are given as follows.

Major points:

Q: Why did not the authors check acetyl-CoA levels after the addition of polyamines to confirm their hypothesis?

RE: Thank you very much for your suggestion. We previously conducted experiments to measure the content of acetyl-CoA with commercial kit. However, we planned to publish this data in the next article, since data provided strong evidence to our hypothesis: on synthetic medium the addition of SPD, but not 1,3-DAP, depletes more than twice the content of Acetyl CoA.

However, you are absolutely right, without these data of acetyl-CoA content the current article looks incomplete. The hypothesis put forward is speculative, not supported by any of the facts. In this regard, we have added a new subsection 2.5. in the Results section containing the new data in Figure 9. In the Materials and Methods section, section 2.7 was added, related to the method for determining cellular acetyl-CoA. A discussion of the added results has been added to the Discussion section. Also, additions in connection with the results obtained were added to the Sections Abstract, Introduction and Conclusion.

Q: Due to the upregulation in the CefD2 expression and the lack of modification in the expression of the cefM transporter, one should expect accumulation of penicillin N inside microbodies. Did authors check this?

RE: Thank you for your valuable question, which reasonably arises from the results obtained.

In order to clearly trace the dynamics of changes, we studied the production of beta-lactams and the level of gene expression every day, starting from 24 h of fermentation (and we also measured the level of gene expression an hour after the start of fermentation, when the production of beta lactams was relatively low). Such consistent observation of the dynamics of beta-lactam production and expression of genes from beta-lactams BGCs provide an opportunity to assess the efficiency of beta-lactam biosynthesis in general. In particular, the increase in cephem production with the addition of polyamines (compared to controls) occurs smoothly and consistently on both media (Figure 5). This may indicate that there is no accumulation of the intermediate in peroxisomes (microbodies). This is not direct, but quite significant indirect evidence.

It should also be noted that the absence of upregulation of all transport genes under the influence of polyamines does not prevent an increase in the final production. In particular, the expression of cefT was practically unchanged by the addition of polyamines during fermentation on both media as shown in Figure 7i for synthetic medium and in Supplementary Materials Figure S3c (Figure S2c, previously) for complex medium. However, an increase in the release of beta-lactams into the culture liquid is observed (Figure 5). In this situation, the question of the accumulation of beta-lactams in the cytoplasm may also arise (biosynthetic genes are upregulated, and the expression of the transporter gene responsible for the release of biosynthesis products into the culture liquid is not changed). However, the experiment shows that the expression of cefT, even without changes in its level, makes it possible to effectively transport the increased flow of beta-lactams. Perhaps this beta-lactam transporter has a certain “margin of safety” for its functioning.

In the case of the work of CefD2 and cefM products in peroxisomes. The total amount of cephems increases with the addition of polyamines during fermentation on both media compared with the control (Figure 5). The last penam in the cephalosporin C biosynthesis pathway, PenN (penicillin N), is the precursor for all cephems in A. chrysogenum (DAOC – deacetoxycephalosporin C; DAC – deacetylcephalosporin C, and CPC) due to activity of CefEF, penN expandase, which carries out the reaction of the expansion of the five-membered ring of penams to the six-membered ring of cephems (Figure 10 – previously Figure 9). 

In our work, in accordance with the methodology, we determined the profile of the beta-lactam fraction from the culture liquid. Therefore, we could only measure PenN, which is secreted into the culture liquid. And its amount was not increased, it was at the level of minor beta-lactam impurities. Since we observe an increase in the content of DAC and CPC, the compounds of beta-lactam biosynthetic pathway from the Next stages after the stages occurring in peroxisomes with the addition of polyamines, it can be assumed that the stages occurring in the peroxisome do not limit the rate of CPC biosynthesis. In current work, we studied the production of beta lactams by adding polyamines on a synthetic medium. However, the same behavior was observed on a complex medium: upregulation of cefD2 [Zhgun A, Eldarov M., Molecules. 2021 Nov; 26(21): 6636 – Figure 7d] against the background of a lack of modification in the expression of the cefM transporter (current article – Supplementary Materials, Figure S3b - Figure S2b, previously).

It is possible that the CefM transporter, like CefT, has a certain “margin of safety” for its functioning, which makes it possible to effectively ensure the transport flow of an increased pool of beta-lactams without changes in the level of expression of the corresponding gene.

The same phenomenon was described for another improved fungal strain Aspergillus terreus, a high-yielding producer of lovastatin. The addition of polyamines led to an increase in the production of lovastatin by 25–45%, upregulation of biosynthetic lov-genes against the background of no changes in the expression of transport gene lovT [Zhgun A, et.al., Appl. Biochem. Microbiol. 2019, 55, 639–648, doi:10.1134/S0003683819060176].

We have added the above material to the Discussion section.

Q: What is the purpose of Supplementary Figure S2? Authors say that it refers to the expression dynamics on synthetic (CP) medium. Is CP medium synthetic? Do authors mean on SN medium? If so, results should be similar to those shown in Figure 7, but some differences can be seen between both figures. Please, clarify this issue.

RE: We thank the reviewer for pointing out our error in the Supplementary Materials section caption, Figure S2 (In the resubmitted version it became Figure S3). We have adjusted this “synthetic (CP) medium” to “complex (CP) medium”. Indeed, this error in the caption led to the question of what this Figure S2 is for, since the expression dynamics of transport genes (cefP, cefM, and cefT) on synthetic (SN) medium is shown in Figure 7. Figure S2 shows previously unpublished data for the expression of transport genes from the beta-lactam BGCs on a complex (CP) medium. Previously, for the fermentation of A. chrysogenum HY on a complex (CP) medium with polyamines, the temporal dynamics of only biosynthetic genes from the beta-lactam BGCs was studied (Zhgun A, Eldarov M., Molecules. 2021 Nov; 26(21): 6636). In the main text of the article, we have shown how the expression of both biosynthetic and transport genes changes over time on a synthetic (SN) medium without polyamines (control) and when they are added (Figure 7). We also shown the relative expression on synthetic (SN)/ complex (CP) media for cef genes (including transport genes) in A. chrysogenum HY. Therefore, data on the level of expression of transport genes on CP media can be calculated from the data in Figures 7 and 8 (the level of expression of transport genes on SN media and relative expression on synthetic (SN)/ complex (CP) media for cef genes). In this regard, we have presented the original data on the level of expression of transport genes on CP media in the Supplementary Materials section. In order to understand the purpose of the Supplementary Materials section, Figure S2, we also referred to this figure in the corresponding section of the manuscript, where we compared the level of expression of transport genes during fermentation on SN and CP media:

“Both for the control SN medium and with the addition of PAs, after 24–48 h and until the end of fermentation, upregulation of all transport genes was observed, by 1.2–2.3 times, compared with the corresponding variants on CP media (Figure 7g, 7h, 7i, and Supplementary Materials Figure S2).”

Minor points:

Q: Line 108. Why authors say "A. chrysogenum HY attenuated strain"? Information about "attenuation" is missing in the text

RE: In the Results section, we used the term "attenuated strain" twice in relation to the A. chrysogenum HY strain:

1) “The level of CPС production in this strain, which became attenuated after multi-round mutagenesis, is extremely sensitive to the medium composition.”

2) “Another phenotypic difference of the A. chrysogenum HY strain is associated with unexpectedly high resistance to ODC inhibitors such as α-difluoromethylornithine (DFMO, or eflornithine) and APA (1-aminooxy-3-aminopropane), which specifically kill A. chrysogenum WT, but not the attenuated A. chrysogenum HY strain”.

However, earlier in the "Introduction" section, we wrote about the decrease in the viability of this strain (improved by classical methods) and cited articles where this phenomenon was described: “Along with a huge increase in the CPC production, this strain has a sharp decrease in viability, which is expressed in slow growth, a reduction in the formation of conidia [37], thinning of the cell wall [38], a decrease in plasma membrane H+-ATPase (PMA, EC: 7.1.2.1) activity and intracellular ATP content [39].”

As a result of your comment, we understand that the reduction in strain viability that occurred as a side event of random mutagenesis is not clearly spelled out. So now we've added a phrase to the "Introduction" section:

“These data show that, as a side event of random mutagenesis, A. chrysogenum HY became attenuated.”

Q: A. chrysogenum should be written in italics throughout the text

RE: corrected to italics throughout the text for “A. chrysogenum”

Q: Some cyrillic characters appear in several parts of the text (e.g. line 358). Please, correct it.

RE: Thank you very much for your comment. We conducted a software screening of the text of the manuscript for the presence of Cyrillic and replaced both individual words and symbols with the Latin alphabet.

Q: Line 431 (446). SN1,3-DAC/CP1,3-DAC. I think authors mean to say SN1,3-DAP/CP1,3-DAP. Please, correct it. 

RE: Corrected to SN1,3-DAP/CP1,3-DAP

Reviewer 2 Report

Addition of poylamines (PAs) increases/triggers CPC production in A. chrysogenum. But some PAs show opposing effects. So far tests only in complex medium containing PA to begin with, which makes it difficult to determine the contribution of single supplements. In the current manuscript the effect of PAs on the production of CPC were tested on a synthetic medium. In this setting, the authors revealed opposing effects of the two PAs 1,3-DAP and SPD on CPC production. While 1,3-DAP led to an increase in CPC production, addition of SPD reduced CPC content (but the direct precursor DAC was increased). Thus, SPD appears to specifically inhibit the final step in the CPC biosynthesis (conversion of DAC into CPC). Expression of the biosynthetic genes involved in CPC biosynthesis was similarly increased after the addition of both PAs, compared to the control (no additive). This is also true for cefG, encoding the DAC-acetyltransferase, involved in converting DAC to CPC. Thus, the authors speculated a difference in the acetyl-CoA pool, as acetyl-CoA is required for the last step. More precisely, the authors hypothesize that the acetyl-CoA gets depleted when SPD is added due to the catabolism of SPD, while this is not the case if 1,3-DAP is added. This in turn results in an increase in DAC/CPC ration. This is the most important statement in this manuscript. Yet, there is not direct proof. Since there seems to be no obvious homolog of the SSAT-encoding gene, I recommend to determine the acetyl-CoA levels in the samples to proof that the reduced rate of conversion of DAC into CPC is indeed due to depleted acetyl-CoA pools in SPD-supplemented samples. There are several possibilities to measure acetyl-CoA e.g., commercially available kits. In my opinion this would validate the hypothesis and thus significantly improve the current manuscript.

Despite from this, the manuscript is well written and clearly structured. I have just a few additional minor comments that would need to be addressed.

Minor comments:

L159 CPA or CPC – I am getting confused with all the abbreviations. Is there any possibility to reduce the number of abbreviations?

L188 Do the authors have an explanation why high nitrogen concentrations (above 50 mM) – irrespective of the source – turned out to be toxic for the fungus? As fungal growth appears not significantly affected at 50 mM, I would like to see fungal growth at concentrations above 50 mM to see the mentioned toxic effect.

Figure 1. Why do the plates look different between WT- and HY-inoculated plates? Does the WT secrete something as the plates appear yellowish? Next, please perform statistical tests to highlight significant changes in hyphal growth. Most of the error bars seem to overlap, so make sure your statistical test makes sense (This is also true for the other bar diagrams). Also, I suggest to use identical y-axis to highlight the growth defect of the HY strain irrespective of the N-source (as it has been done e.g., for Figure 2).

L219. Supplementation of sodium nitrate (50 mM) does not appear to be toxic in Figure 1. Growth appears similar to L-Gln, which is mentioned to be good growth supplement. The same is true for urea. Please, either explain or re-write.

Figure 2. Please correct the legend in b and c. It is identical with Figure 1 but should read 1, 5 and 10 mM, I guess. Otherwise L232ff would have to be corrected.

L257 (and throughout the manuscript) Organism in italics

L260f production cannot decrease rather go for increase was slowed down compared to…

Figure 3b. I’ll have to question the significance of the obtained results since the error bars are clearly overlapping for almost all bars. In case of overlapping error bars (given the sample sizes are equal) the P value must be greater than 0.05, so this is not statistically significant. If there is something I am missing here, please elaborate.

L279f. Addition of 5 mM SPD is apparently not toxic as shown in Figure 2.

Figure 6. see comment on Figure 3, especially when the authors point out that the ratio of DAC/CPC is not different when grown in CP.

Figure 7. see comment on Figure 3. Especially for g-h, for which the authors state that there is no significant difference (see L422ff.).

The authors mention in L 143f that there is no significant change in cefG expression, encoding the enzyme involved in converting DAC into CPC. However, in figure 7f there is a visible significant different in the expression of cefG between DAP- and SPD-supplemented samples. Which, however, theoretically should result in higher CPC levels in SPD-treated samples, which obviously is not the case, which is also stated by the authors in L408ff. This needs to be addressed by the authors. Also, how do the authors explain that transport proteins are not overproduced upon biosynthesis stimulation, even when higher levels are detected outside of the cell?

L504 Please define CN medium, identical to SN?

Please provide a more detailed material and methods section e.g., HPLC analysis. What equipment what used, describe the exact constitution of the mobile phase, … What where the exact conditions during fermentation, …

Author Response

Responses to Reviewer #2

Question: Addition of polyamines (PAs) increases/triggers CPC production in A. chrysogenum. But some PAs show opposing effects. So far tests only in complex medium containing PA to begin with, which makes it difficult to determine the contribution of single supplements. In the current manuscript the effect of PAs on the production of CPC were tested on a synthetic medium. In this setting, the authors revealed opposing effects of the two PAs 1,3-DAP and SPD on CPC production. While 1,3-DAP led to an increase in CPC production, addition of SPD reduced CPC content (but the direct precursor DAC was increased). Thus, SPD appears to specifically inhibit the final step in the CPC biosynthesis (conversion of DAC into CPC). Expression of the biosynthetic genes involved in CPC biosynthesis was similarly increased after the addition of both PAs, compared to the control (no additive). This is also true for cefG, encoding the DAC-acetyltransferase, involved in converting DAC to CPC.

Response: We are grateful to the reviewer for the careful and critical analysis of the manuscript and valuable comments. We have carefully revised the manuscript in accordance with your suggestions, responded to all questions and made adjustments and corrections to the indicated places in accordance with your recommendations. In the revised manuscript, we have marked the changes in yellow, and the answers are given point by point as follows.

Q: Thus, the authors speculated a difference in the acetyl-CoA pool, as acetyl-CoA is required for the last step. More precisely, the authors hypothesize that the acetyl-CoA gets depleted when SPD is added due to the catabolism of SPD, while this is not the case if 1,3-DAP is added. This in turn results in an increase in DAC/CPC ration. This is the most important statement in this manuscript. Yet, there is not direct proof. Since there seems to be no obvious homolog of the SSAT-encoding gene, I recommend to determine the acetyl-CoA levels in the samples to proof that the reduced rate of conversion of DAC into CPC is indeed due to depleted acetyl-CoA pools in SPD-supplemented samples. There are several possibilities to measure acetyl-CoA e.g., commercially available kits. In my opinion this would validate the hypothesis and thus significantly improve the current manuscript.

Despite from this, the manuscript is well written and clearly structured. I have just a few additional minor comments that would need to be addressed.

RE: Thank you very much for your suggestion. We previously conducted experiments to measure the content of acetyl-CoA with commercial kit. However, we planned to publish this data in the next article, since data provided strong evidence to our hypothesis: on synthetic medium the addition of SPD, but not 1,3-DAP, depletes more than twice the content of Acetyl CoA.

However, you are absolutely right, without these data of acetyl-CoA content the current article looks incomplete. The hypothesis put forward is speculative, not supported by any of the facts. In this regard, we have added a new subsection 2.5. in the Results section containing the new data in Figure 9. In the Materials and Methods section, section 2.7 was added, related to the method for determining cellular acetyl-CoA. A discussion of the added results has been added to the Discussion section. Also, additions in connection with the results obtained were added to the Sections Abstract, Introduction and Conclusion.

Minor comments:

Q: L159 CPA or CPC – I am getting confused with all the abbreviations. Is there any possibility to reduce the number of abbreviations?

RE: corrected CPA to CPC (cephalosporin C). We made a mistake here, thanks for pointing it out. To ensure that there are no misunderstandings when reading the article, we have added to the Abbreviations section, which follows immediately before the literature section, transcripts for the abbreviations commonly used in the article: CPA - Agarized complex (medium) and CP - Complex (medium). We carefully checked the entire text for the use of abbreviations and found another mistake, where instead of CPC (cephalosporin C) the abbreviation CPA was used; also fixed.

In accordance with your recommendation, we have reduced the number of abbreviations used in the text - in all situations where this is possible. We removed the entered abbreviations and replaced them with full words for: IPN - Isopenicillin N, PenN - Penicillin N, PUT - Putrescine, and SPM - Spermine. As for other abbreviations used in our article, when describing the results obtained, if the introduced abbreviations of key beta-lactam metabolites, polyamines and culture media are not used, the sentences become longer, which can make it difficult to perceive the results. That is why we introduced the abbreviations used.

Q: L188 Do the authors have an explanation why high nitrogen concentrations (above 50 mM) – irrespective of the source – turned out to be toxic for the fungus? As fungal growth appears not significantly affected at 50 mM, I would like to see fungal growth at concentrations above 50 mM to see the mentioned toxic effect.

RE: Thanks for your question. In addition to the indicated concentrations, we measured concentrations of 100 mM (Additional data with this concentration are given in the new added Supplementary Materials Figure S2). In this regard, our phrase “all compounds above 50 mM turned out to be toxic” is incorrect. We have replaced it with “concentrations of 100 mM or more for all compounds turned out to be toxic”. If we did more painstaking work, then we can certainly find a certain concentration up to 5-10 mm, toxic for each of the studied strains. For example, for NH4CL it is in the range of 10-50 mM, for others it is in the range of 50-100 mM. We did not do this work, however, the selected concentration of 50 mM L-asparagine proved to be an effective component of the medium, on which it was possible to obtain the opposite effect from the addition of the studied polyamines.

We do not have a clear explanation why concentrations above 100 mM of the studied nitrogen sources are toxic for A. chrysogenum strains (as components of Czapek's agar medium). Perhaps this is due to the peculiarities of the metabolism of a particular organism. For example, for a number of strains of Aspergillus terreus studied by us on a 100 mM synthetic medium, the concentrations of some nitrogen sources are nontoxic. Toxicity appears after 200 mM.

Q: Figure 1. Why do the plates look different between WT- and HY-inoculated plates? Does the WT secrete something as the plates appear yellowish?

RE: Thank you for noticing this moment. Previously we described these phenotypic differences between A. chrysogenum wild-type strain and its highly active counterpart [1]:

“Colonies of A. chrysogenum WT have a characteristic yellow-cream color, which develops after 5–10 days, depending on the cultivation conditions on the agar nutrient medium, and due to the biosynthesis of the SM (secondary metabolite) sorbicillin [2].

Targeted knockout of the sorA or sorB genes (encoding the central enzymes, polyketide synthases of sorbicilinoid BGC) in the A. chrysogenum wild-type strain results in a white phenotype, exactly the same as in A. chrysogenum HY [2]. It can be assumed that in A. chrysogenum HY, the biosynthesis of sorbicillin was disrupted; therefore, the yellow-cream color does not develop either during the idiophase period or under the influence of polyamines. This may be due to screening after random mutagenesis when mutants with disruption of the biosynthesis of alternative SM are also selected [3]. The inability of the improved strains to synthesize some alternative SM was shown at the molecular level and may be due to the release of additional resources for targeted biosynthesis [3]. Disruption of the biosynthesis of polyketide sorbicillin could lead to an additional increase in the production of cephalosporin C in A. chrysogenum HY”.

  1. Zhgun, A.A.; Eldarov, M.A. Polyamines Upregulate Cephalosporin C Production and Expression of β-Lactam Biosynthetic Genes in High-Yielding Acremonium chrysogenum Strain. Mol. 2021, Vol. 26, Page 6636 2021, 26, 6636, doi:10.3390/MOLECULES26216636.
  2. Chen, G.; Chu, J. Characterization of Two Polyketide Synthases Involved in Sorbicillinoid Biosynthesis by Acremonium chrysogenum Using the CRISPR/Cas9 System. Appl. Biochem. Biotechnol. 2019, 188, 1134–1144, doi:10.1007/s12010-019-02960-z.
  3. Terfehr, D.; Dahlmann, T.A.; Kück, U. Transcriptome analysis of the two unrelated fungal β-lactam producers Acremonium chrysogenum and Penicillium chrysogenum: Velvet-regulated genes are major targets during conventional strain improvement programs. BMC Genomics 2017, 18, 272, doi:10.1186/s12864-017-3663-0.

            We have added the necessary phrase to the Introduction section explaining why the phenotypes of the HY and WT strains shown in Figures 1 and 2 are different:

“This HY strain also lost a characteristic yellow-cream color, which develops in the WT strain after 5–10 days, depending on the cultivation conditions on the agarized nutrient medium, and due to the biosynthesis of the secondary metabolite sorbicillin” 

Q: Next, please perform statistical tests to highlight significant changes in hyphal growth. Most of the error bars seem to overlap, so make sure your statistical test makes sense (This is also true for the other bar diagrams). Also, I suggest to use identical y-axis to highlight the growth defect of the HY strain irrespective of the N-source (as it has been done e.g., for Figure 2).

RE: According to your recommendations we perform statistical tests to highlight significant changes in hyphal growth and use identical y-axis to highlight the growth defect of the HY strain irrespective of the N-source. Corrections are made to the updated Figure 1.

Q: L219. Supplementation of sodium nitrate (50 mM) does not appear to be toxic in Figure 1. Growth appears similar to L-Gln, which is mentioned to be good growth supplement. The same is true for urea. Please, either explain or re-write.

RE: Thank you for noticing this description of toxicity. We re-write this phrase, excluding as toxic agents: for the wild-type strain - 50 mM sodium nitrate and for the HY strain 50 mM urea. The toxicity of 50 mM urea (as well as 10-50 mM NH4Cl) for WT strain in addition to a decrease in growth is expressed in the disappearance of the characteristic yellow-cream pigment color, which indicates a violation of sorbicillin biosynthesis [1, 2]. In this regard, we have added one more phrase: “At this concentration of NH4Cl or urea, the production of secondary metabolite sorbicillin was also disrupted in the WT strain, which was expressed in the absence of the characteristic yellow-cream color.”

  1. Chen, G.; Chu, J. Characterization of Two Polyketide Synthases Involved in Sorbicillinoid Biosynthesis by Acremonium chrysogenum Using the CRISPR/Cas9 System. Appl. Biochem. Biotechnol. 2019, 188, 1134–1144, doi:10.1007/s12010-019-02960-z.
  2. Zhgun, A.A.; Eldarov, M.A. Polyamines Upregulate Cephalosporin C Production and Expression of β-Lactam Biosynthetic Genes in High-Yielding Acremonium chrysogenum Strain. Mol. 2021, Vol. 26, Page 6636 2021, 26, 6636, doi:10.3390/MOLECULES26216636.

Q: Figure 2. Please correct the legend in b and c. It is identical with Figure 1 but should read 1, 5 and 10 mM, I guess. Otherwise L232ff would have to be corrected.

RE: Thank you for your comment. You are absolutely correct in pointing out our mistake. We have corrected our mistake. The legend in b and c conflicted not only with L232ff, but also with the original visual data and the legend in a.

Q: L257 (and throughout the manuscript) Organism in italics

RE: We have fixed the italicization of the Organism throughout the text

Q: L260f production cannot decrease rather go for increase was slowed down compared to…

RE: corrected “decreased” to “was slowed down”

Q: Figure 3b. I’ll have to question the significance of the obtained results since the error bars are clearly overlapping for almost all bars. In case of overlapping error bars (given the sample sizes are equal) the P value must be greater than 0.05, so this is not statistically significant. If there is something I am missing here, please elaborate.

RE: Thank you for your comment. We have redone the statistical analysis. Indeed, for most of the results shown in Figure 3b, P value turned out to be greater than 0.05, so this is not statistically significant. This was our error in the statistical processing of the data. We have also redone the statistical analysis for other histograms in the manuscript, and changes have been made to the corresponding figures.

Q: L279f. Addition of 5 mM SPD is apparently not toxic as shown in Figure 2.

RE: Thanks for this comment. Figure 2 shows preliminary data for agarized Czapek–L-Asn medium. Under these conditions, the addition of 5 mM polyamines was not toxic (but increasing the concentration of polyamines to 10 mM had a toxic effect). Phrase L279f emphasizes that 5 mM polyamines are also non-toxic on a liquid synthetic medium. This fact is not quite obvious a priori, since agar itself, as a component of the medium, could partially remove the toxic effect of polyamines.

In order to connect the data obtained in the work on the absence of toxicity when adding 5 mM polyamines to agar and liquid media, we added the phrase to this section of the article: “This is also consistent with preliminary data obtained on the agarized Czapek-L-Asn medium, where the addition of 5 mM PAs was also non-toxic (Figure 2).”   

Q: Figure 6. see comment on Figure 3, especially when the authors point out that the ratio of DAC/CPC is not different when grown in CP.

RE: Recalculated. Corrections are made in Figure 6.

Figure 7. see comment on Figure 3. Especially for g-h, for which the authors state that there is no significant difference (see L422ff.).

RE: Recalculated. Corrections are made in Figure 7.

Q: The authors mention in L 143f that there is no significant change in cefG expression, encoding the enzyme involved in converting DAC into CPC. However, in figure 7f there is a visible significant different in the expression of cefG between DAP- and SPD-supplemented samples. Which, however, theoretically should result in higher CPC levels in SPD-treated samples, which obviously is not the case, which is also stated by the authors in L408ff. This needs to be addressed by the authors. Also, how do the authors explain that transport proteins are not overproduced upon biosynthesis stimulation, even when higher levels are detected outside of the cell?

RE: Thank you for your question. Indeed, we have obtained a phenomenon that is reproducible on both nutrient media: When polyamines are added, the total production of cephems increases (Figure 5), which is expressed in the upregulation of biosynthetic genes against the background of the absence of transport gene expression in the change. In this regard, a detailed discussion of these results was added to the Discussion section:

“The absence of upregulation of transport genes (cefP, cefM, and cefT) against the background of upregulation of biosynthetic genes with the addition of PAs does not prevent an increase in the final production of cephems (Figures 5, 7). In particular, the expression of cefT was practically unchanged by the addition of PAs during fermentation on both media (Figure 7i, Supplementary Materials Figure S3c). However, an increase in the release of major cephems, DAC and CPC, into the culture liquid was observed (Figure 5). Perhaps this beta-lactam transporter has some “safety margin” for its functioning, or cephems may be released from A. chrysogenum cells in some other way, as discussed earlier [50]. On the other hand, the lack of modification in the expression of the cefM transporter and upregulation in the CefD2 expression should expect accumulation of penicillin N inside peroxisomes. Such an effect of PAs on the expression of these genes is observed both in SN medium (Figures 7d, 7h) and in CP medium ([11], Supplementary Materials Figure S3b). In our work, in accordance with the methodology, the profile of the beta-lactam fraction was determined from the culture liquid, and the content of beta-lactams in peroxisomes was not measured separately. Therefore, we could only measure penicillin N, which is secreted into the culture liquid. And its amount was not increased; it was at the level of minor beta-lactam impurities (Figure 4). However in order to clearly trace the dynamics of changes, we studied the production of beta-lactams and the level of gene expression every day, starting from 24 h of fermentation; we also measured the level of gene expression an hour after the start of fermentation, when the production of beta lactams was relatively low (Figures 3, 7, Supplementary Materials Figure S3). Such consistent observation of the dynamics of beta-lactam production and expression of genes from beta-lactams BGCs provide an opportunity to assess the efficiency of beta-lactam biosynthesis in general. Since we observe an increase in the content of DAC and CPC, the compounds from next stages of beta-lactam biosynthetic pathway, with the addition of PAs, it can be assumed that there is no accumulation of the intermediate penicillin N in peroxisomes. This is not direct, but quite significant indirect evidence. Perhaps the CefM transporter, like CefT, has a certain “margin of safety” for its functioning, which makes it possible to effectively ensure the transport flow of an increased pool of beta-lactams without changing the level of expression of the corresponding gene. The same phenomenon was described for another improved fungal strain Aspergillus terreus, a high-yielding producer of lovastatin. The addition of PAs led to an increase in the production of lovastatin (by 25–45%) and upregulation of biosynthetic lov-genes against the background of no changes in the expression of transport gene lovT [42].”

Q: L504 (521) Please define CN medium, identical to SN?

RE: Thank you for your comment. We mistakenly wrote instead of SN (synthetic) CN medium. We corrected CN medium to SN medium in the indicated place L504 (521), as well as in some other places in the text where we made the same mistake in the designation of the synthetic medium

Q: Please provide a more detailed material and methods section e.g., HPLC analysis. What equipment what used, describe the exact constitution of the mobile phase, … What where the exact conditions during fermentation, …

RE: We have added more detailed information about how the HPLC analysis was performed in the corresponding section 4.6. In particular, we indicated that we used the equipment the Agilent 1200 liquid chromatograph (Agilent Technologies Inc., Santa Clara, CA, USA), described the exact composition of the mobile phase. As for the fermentation conditions, they are detailed in section 4.4., which we referred to in this paragraph of the methodology, talking about how the samples were obtained. We also added how we prepared selected samples for HPLC “centrifuged in a 1,5 ml Eppendorf at 16000 g, 10 min”, used for HPLC analysis 10 μl of supernatant).

Round 2

Reviewer 1 Report

Authors have answered to all my questions and have carried out the experiments requested. Therefore, I find the article suitable for publication.